# Immunopeptidomic Analysis of BoLA-I and BoLA-DR Presented Peptides from *Theileria parva* Infected Cells

**DOI:** 10.3390/vaccines10111907

**Published:** 2022-11-11

**Authors:** Timothy Connelley, Annalisa Nicastri, Tara Sheldrake, Christina Vrettou, Andressa Fisch, Birkir Reynisson, Soren Buus, Adrian Hill, Ivan Morrison, Morten Nielsen, Nicola Ternette

**Affiliations:** 1The Roslin Institute, The Royal (Dick) School of Veterinary Science, The University of Edinburgh, Edinburgh EH25 9RG, UK; 2The Jenner Institute, Nuffield Department of Medicine, The University of Oxford, Oxford OX3 7BN, UK; 3Ribeirão Preto College of Nursing, University of São Paulo, Av Bandeirantes, Ribeirão Preto 3900, Brazil; 4Department of Health Technology, Technical University of Denmark, DK-2800 Copenhagen, Denmark; 5Laboratory of Experimental Immunology, Department of Immunology and Microbiology, Faculty of Health and Medical Sciences, University of Copenhagen, DK-2200 Copenhagen, Denmark; 6Instituto de Investigaciones Biotecnológicas, Universidad Nacional de San Martín, San Martín CP1650, Argentina

**Keywords:** cattle, parasite-protozoan, MHC, antigen processing and presentation

## Abstract

The apicomplexan parasite *Theileria parva* is the causative agent of East Coast fever, usually a fatal disease for cattle, which is prevalent in large areas of eastern, central, and southern Africa. Protective immunity against *T. parva* is mediated by CD8^+^ T cells, with CD4^+^ T-cells thought to be important in facilitating the full maturation and development of the CD8^+^ T-cell response. *T. parva* has a large proteome, with >4000 protein-coding genes, making T-cell antigen identification using conventional screening approaches laborious and expensive. To date, only a limited number of T-cell antigens have been described. Novel approaches for identifying candidate antigens for *T. parva* are required to replace and/or complement those currently employed. In this study, we report on the use of immunopeptidomics to study the repertoire of *T. parva* peptides presented by both BoLA-I and BoLA-DR molecules on infected cells. The study reports on peptides identified from the analysis of 13 BoLA-I and 6 BoLA-DR datasets covering a range of different BoLA genotypes. This represents the most comprehensive immunopeptidomic dataset available for any eukaryotic pathogen to date. Examination of the immunopeptidome data suggested the presence of a large number of coprecipitated and non-MHC-binding peptides. As part of the work, a pipeline to curate the datasets to remove these peptides was developed and used to generate a final list of 74 BoLA-I and 15 BoLA-DR-presented peptides. Together, the data demonstrated the utility of immunopeptidomics as a method to identify novel T-cell antigens for *T. parva* and the importance of careful curation and the application of high-quality immunoinformatics to parse the data generated.

## 1. Introduction

A major challenge to the development of novel vaccines for complex intracellular pathogens is the identification of relevant T-cell antigens. One example of this is *Theileria parva*, the causative agent of East Coast fever (ECF), a highly pathogenic disease for cattle that is prevalent in large areas of eastern, central, and southern Africa. ECF is estimated to kill ~1 million cattle a year and inflict an annual economic cost of up to 600 million USD [1]; as a major proportion of this burden is borne by smallholder farmers, ECF poses a major threat to the livelihoods and food security of some of the poorest communities in the world. *T. parva* is transmitted by the brown-eared tick (*Rhipicephalus appendiculatus*), which deposits sporozoites into the skin of the cattle host while taking a blood meal. The sporozoites rapidly invade host lymphocytes and, once within cells, transition to a schizont form. Schizont-infected cells undergo transformation and are induced into a phase of sustained proliferation, during which infection with the parasite is maintained in the ‘daughter’ cells; dissemination of these cells and the host’s inflammatory response ultimately lead to the clinical symptoms of the disease, and frequently the death of the infected animal. 

Currently, ECF control options are limited to the intensive use of acaricides, a single theilericidal chemotherapeutic agent, and an ‘infection-and-treatment’ (ITM) form of immunisation, which are used either alone or in combination. However, none of these options is optimal or sustainable in the longer term due to, among other reasons, increasing acaricide resistance, the expense and poor efficacy of the theilericidal agent, and the logistical issues of deploying a live vaccine, which requires an extensive liquid nitrogen cold chain, in rural Africa. Thus, alternative forms of control are actively being sought. Predominant among these is the development of novel subunit vaccines. Studies on the immune responses generated by ITM immunisation have demonstrated that protection is associated with major histocompatibility complex class I (MHCI)-restricted CD8^+^ T-cell responses [2,3,4,5]. Thus, CD8^+^ T-cell antigens are a major target for the development of efficacious subunit vaccines. The role of CD4^+^ T-cells in promoting the induction and maintenance of functionally mature CD8^+^ T-cells suggests that antigens presented by BoLA class II molecules (BoLA-II) will also be required for an effective CD8^+^ T-cell-targeting vaccine [6,7]. Selection of CD8^+^/CD4^+^ T-cell antigens for inclusion in a vaccine will require consideration of both the diversity of the MHCI and the MHCII genotypes in the target cattle populations and the antigenic variability between different *T. parva* strains [8,9,10,11]. Consequently, it is likely that the repertoire of antigens needed for an effective subunit vaccine will be complex and will need to be minimised by careful selection, if the production of the vaccine is to be practicable. 

Intensive efforts applying conventional antigen-screening approaches (using *T. parva* cDNA and/or peptide libraries) to identify candidate antigens recognised by *T. parva*-specific T-cells from immune animals have resulted in the identification of a limited number of CD8^+^ T-cell antigens (Tp1-11 and Tp32-35) from the Muguga genome-reference strain of *T. parva* [12,13]. Recent work has also identified 26 antigens containing BoLA-DR and/or BoLA-DQ epitopes recognised by *T. parva* Muguga-specific CD4^+^ T-cells [13]. However, for many cattle MHC genotypes, there are still no identified *T. parva* antigens, and due to the diversity of both the bovine MHC and the parasite variants present ‘in the field’; these epitopes would be anticipated to only provide protection to a proportion of the cattle populations and only against a subset of *T. parva* strains. The expression of >4000 proteins by *T. parva* schizonts [14] places severe logistical limitations on the capacity for such approaches to screen the full *T. parva* proteome across multiple strains and MHC haplotypes that would be required to generate a comprehensive list of antigens. Such efforts are further impeded by the marked immunodominance that is characteristic of the CD8^+^ T-cell responses against *T. parva* [15], which narrowly restricts the number of epitopes that elicit detectable T-cell responses in individual animals following natural infection or ITM immunisation. Many of the immunodominant epitopes exhibit high levels of strain polymorphism, limiting their potential utility as future candidate antigens [11]. As such, there is a need to complement ongoing antigen identification efforts with alternative strategies. 

Recent advances in nanoflow ultra-high performance liquid chromatography coupled to mass-spectrometry (LC-MS) used in peptidomics have significantly increased the sensitivity, dynamic range, and mass accuracy of the technique, such that the analysis of peptides is orders of magnitude superior to that possible a decade ago. Application of this technique in peptide-MHC (pMHC) elution studies has greatly enhanced our understanding of the repertoire of MHC-bound peptides (‘immunopeptidome’) under steady-state conditions and following neoplastic transformation (for just a few recent examples, see references [16,17,18]). Sequencing of MHC-eluted peptides by LC-MS has also been used to examine the pathogen-derived peptides presented by MHC molecules for a number of infectious agents, including the Mumps virus, HIV, *Toxoplasma*, *Plasmodium*, *Mycobacterium*, and *Leishmania* [19,20,21,22,23,24,25]. Data from these studies have identified a number of proteins that could be used to generate immune responses when administered as a component of an experimental vaccine. 

Herein, we report on an immunopeptidomic analysis of *T. parva*-derived peptides presented by infected cells and demonstrate how these mass-spectrometry data, combined with immuno-informatics, can be used as an alternative approach to formulate a list of new *T. parva* candidate antigens. To the authors’ knowledge, this is the first time immunopeptidomics has been used to study a bovine pathogen, and the first time the immunopeptidome of a protozoan pathogen has been studied in the context of multiple MHCI and MHCII haplotypes. We identified a total of 74 and 15 unique BoLA-I and BoLA-DR-restricted peptides—demonstrating the capacity to use this approach to rapidly characterise the immunopeptidome of *T. parva*-infected cells and so identify novel candidate CD8^+^ and CD4^+^ T-cell antigens.

## 2. Materials and Methods

### 2.1. BoLA-Defined Cell Lines

Cell lines were established from animals for which BoLA-I and BoLA-DR genotypes had been determined by a combination of conventional Sanger-based sequencing of amplicons generated using MHCI-allele-specific PCR [26], Sanger-based typing of BoLA-DR [27], and MiSeq-based BoLA-I and BoLA-DR sequencing [28]. *Theileria parva*-infected B- and T-cell lines were generated and characterised as part of previous studies and maintained using well-established protocols [29]. The cell lines had all been generated from Holstein-Friesian animals from the University of Edinburgh Herd and had been used under a license granted under the UK Animal (Scientific Procedures) Act 1986.

### 2.2. Peptide-BoLA-I and Peptide-BoLA-DR Complex Purification

Cells were harvested whilst in a log growth phase, and trypan blue staining was used to verify that >95% of the cells were viable at the point of harvest. Cells were washed twice with ice-cold PBS and then lysed in buffer (1% IGEPAL, 15 mM TRIS pH 8.0, 300 mM NaCl, and a complete protease inhibitor (Roche, Welwyn Garden City, UK)) at a density of 2 × 10^8^ cells/mL for 1 min, diluted with PBS 1:1 and solubilized for 45 min at 4 °C. Lysates were cleared with two-step centrifugation at 500× *g* for 15 min, followed by 15,000× *g* for 45 min at 4 °C. pBoLA-I complexes were captured directly from the lysate using a pan-specific anti-BoLA-I antibody (ILA-88) covalently conjugated to protein A sepharose immunoresin (Amintra, Expedeon, Cambridge, UK) at a concentration of 5 mg/mL. BoLA-DR complexes were captured from the lysate, following a preliminary removal of pBoLA-I complexes (using BoLA-I capture as described above), using a pan-specific anti-BoLA-DR antibody (ILA-21) conjugated to protein A sepharose immunoresin at a concentration of 5 mg/mL. Captured pBoLA-I and pBoLA-DR complexes were washed sequentially using buffers of 50 mM Tris buffer, pH 8.0 containing 150 mM NaCl, then 400 mM NaCl, and finally 0 mM NaCl, prior to elution of the BoLA-bound peptides, BoLA protein chains, and β2M in 10% acetic acid and stored as described previously [30].

### 2.3. High Performance Liquid Chromatography (HPLC) Fractionation

Affinity column-eluted material was resuspended in 120 μL loading buffer (0.1% formic acid, 1% acetonitrile in water) and loaded onto a 4.6 × 50 mm ProSwiftTM RP-1S column (Thermo Scientific, Waltham, MA, USA) for reverse-phase chromatography on an Ultimate 3000 HPLC system (Thermo Scientific). Elution was performed using a 0.5 mL/min flow rate for over 5 min on a gradient of 2–35% buffer B (0.1% formic acid in acetonitrile) in buffer A (0.1% formic acid). Eluted fractions were collected from 1 to 8.5 min, for 30 s each. Protein detection was performed at 280 nm. Even and odd eluted fractions were pooled together, vacuum, dried, and stored at −80 °C until use.

### 2.4. LC-MS^2^ Analysis

Samples were suspended in a 20 μL loading buffer and analysed on an Ultimate 3000 nano UPLC system online coupled to either an Orbitrap Fusion^Tm^ Tribrid^Tm^ Mass Spectrometer or a Q Exactive™ HF-X Hybrid Quadrupole-Orbitrap™ Mass Spectrometer (Thermo Scientific). Peptides were separated on a 75 μm × 50 cm PepMap C18 column using a 1 or 2 h linear gradient from 2–5% buffer A to 35% buffer B at a flow rate of 250 nL/min (approx. 600 bar at 40 °C). Peptides were introduced into the mass spectrometer using a nano Easy Spray source (Thermo Scientific) at 2000 V. The ion transfer tube temperature was set to either 305 °C (Fusion Lumos), or 250 °C (HF-X). Subsequent isolation and higher-energy C-trap dissociation (HCD) were induced on the 20 most abundant ions per full MS scan with an accumulation time of 128 ms and an isolation width of 1.2 Da (Fusion Lumos) or 1.6 Da (HF-X). All fragmented precursor ions were actively excluded from repeated selection for 8 s (Fusion) or 15 s (HF-X). The mass spectrometry proteomics data have been deposited to the ProteomeXchange Consortium via the PRIDE partner repository [31] with the dataset identifiers PXD008151 and PXD024053.

### 2.5. MS Data Analysis

The sequence interpretations of mass spectrometry spectra were performed using a database containing all bovine UniProt entries (total of 41,610 entries) and 4084 entries for the *T. parva* Muguga proteome [14]. The spectral interpretation was performed using the novo-assisted database search with PEAKS v8.5 or 10 (Bioinformatics Solutions), in ‘no enzyme’ mode, with mass tolerances of 5 ppm for precursor ions and 0.03 Da for fragment ions. The data were further searched against 313 in-build peptide modifications. To be included in the downstream analyses, *T. parva* peptides had to meet the following criteria: (1) a peptide-spectrum matching score (−10 lgP) of >20, (2) no predicted post-translational modifications, (3) a minimum of 2 amino acids difference from any bovine peptide sequence, and (4) for BoLA-DR peptides, predicted to not be a BoLA-I-binding peptide (the latter due to evidence of coprecipitation of BoLA-I peptides in the BoLA-DR eluted datasets [32]). Prediction of the MHC binding capacity of *T. parva* peptides was conducted using NetMHCpan-4.1 [33] for BoLA-I, and NetBoLA-IIpan [32] for BoLA-DR eluted peptides. Following the default parameters, peptides with a percent rank predicted score of >20% were considered to be nonbinders, whilst peptides with scores <2% and <5% were considered to be binders for BoLA-I and BoLA-DR, respectively. Coprecipitants were identified as overlapping peptides present in multiple cell-lines carrying nonhomologous BoLA-I/BoLA-DR molecules, where the lowest percent rank prediction score in at least one sample was >20% (i.e., predicted to be a nonbinder).

### 2.6. IFNG ELISPOT

IFNG ELISPOT was conducted using a standard format. In brief, a capture anti-IFNG monoclonal antibody (CC330—Biorad, Watford, UK) was bound to a pre-wetted PVF membrane multiscreen plate (Millipore, Watford, UK). Autologous *T. annulata* cells (used as antigen-presented cells −2 × 10^4^ cells per well) were loaded with peptides at a concentration of 5 μg/mL for 2 h. At 37 °C before CD8^+^ T-cells were added at a density of 1 × 10^4^ cells/well. Plates were incubated at 37 °C for 20 h before washing and identification of IFNG-producing cells by the addition of a biotinylated detection anti-IFNG antibody (CC302b—produced in-house) and development by sequential use of Vectastain (Vector Laboratories, Burlingame, CA, USA) and AEC substrate (Calbiochem, Watford, UK) solutions. Analysis of spot-forming units (SFU) was completed using an AID automated ELISpot reader (AID, Strassberg, Germany).

### 2.7. In Vitro Measurement of Peptide-BoLA-I Binding

The extracellular domains (positions 1–275, i.e., truncated at the transmembrane region) of BoLA-I heavy chain molecules had previously been produced as recombinant proteins and used to measure the affinity of peptide-BoLA-I interactions using human beta-2-microglubulin as the light chain component [34,35]. A previously described assay measuring peptide-human MHCI dissociation rate at 37 °C was adapted to these BoLA-I molecules [36]. Briefly, this assay used the dissociation of the invariant beta-2-microglubulin as a proxy to measure the dissociation of the peptides offered to the BoLA-I. The dissociation at 37 °C of ^125^I-radiolabeled beta-2-microglubulin was monitored in real-time by a high-throughput scintillation proximity assay, and the half-life of the dissociation was determined. Peptides that failed to bind to BoLA-I molecules did not register a half-life period; the length of the half-life was used to infer the stability and binding strength of individual peptide-BoLA-I complexes.

## 3. Results

### 3.1. Identification of BoLA-I Associated Peptides Derived from T. parva

To assess the ability to detect *T. parva*-derived BoLA-I-associated peptides using an immunopeptidomics approach, BoLA-I-associated peptides were purified with three *T. parva*-infected cell lines (TP), and the peptide fractions were analysed by LC-MS. Each TP was homozygous for a different MHC haplotype, which expressed 1, 2, or 4 BoLA-I genes (in contrast to humans and mice, there is a variable number of MHCI loci expressed in different BoLA-I haplotypes, ranging from 1 to 4 [5,37]; 641TP (A18: 6:01301), 1011TP (A10: 2:01201, 3:00201), and 2229TP (A14: 2:02501, 4:02401, 1:02301, 6:04001). From these cell lines, a total of 7672, 6961, and 6871 peptide sequences were identified, respectively (Figure 1A). Due to the high degree of homology between the sections of the bovine and *T. parva* genomes, a blast search of all peptide sequences putatively derived from *T. parva* against the bovine proteome was completed. To remove peptides potentially derived from bovine protein, variants from further analysis sequences that were less than two amino acids (a.a.) different from a matching sequence in the bovine proteome were excluded. After filtering, 25 (0.32%), 18 (0.25%), and 25 (0.36%) peptide sequences were identified as being unambiguously derived from *T. parva* proteins in the 1011TP, 2229TP, and 641TP cell lines, respectively (total number and number of unique *T. parva* peptides identified were 68 and 62, respectively—Table 1, Figure 1A).

To verify the accuracy of the MS spectral sequence annotation of the identified *T. parva* peptides, we employed a spectral matching approach, performing analysis of synthetic peptides under identical LC-MS conditions for a subset of the peptides (*n* = 33). The spectra obtained from the synthetic peptides matched those measured for the majority of the peptides analysed (*n* = 31, 93.9%), confirming their correct identification (Table 1, Figure 1B). 

The capacity of the identified peptides to bind to the BoLA-I molecules expressed in the cell lines was predicted using the NetMHCpan4.1 algorithm [38], with peptides achieving a percent predicted rank binding score of <2% considered to be BoLA-I binders (Table 1). Only 33.8% of the *T. parva* peptides were predicted to be BoLA-I binders, in contrast to the bovine-derived peptides from the same samples, where ~90% of peptides were predicted to be capable of binding to a BoLA-I molecule expressed in the respective samples (Figure 1C). 

Notably, unlike the bovine-derived peptides, the *T. parva* peptides did not exhibit a Gaussian (‘normal’) length distribution, and the low percentage of *T. parva* peptides predicted to be BoLA-I binders appeared to be due, in part, to the presence of a substantial fraction of peptides (*n* = 32, 47%) that were longer than the canonical 8–12 a.a. length of MHCI-binding peptides (Figure 1D). Only 9.4% of ≥ 13 mer peptides (*n* = 3/32) were predicted to be BoLA-I binders, whereas 55.6% (*n* = 20/36) of the 8–12 mer peptides had a predicted rank binding score of <2% (Table 1). This overrepresentation of longer peptides in the *T. parva*-derived peptidome suggested either a specific property for parasite peptides associated with MHC-I complexes, as has previously been observed for *Toxoplasma gondii* peptides [20], or a high proportion of coprecipitating peptides in the pathogen-derived fraction of the peptidome. The derivation of 13/32 (~40%) of ≥13 mer *T. parva* peptides from a single 28 a.a. region of one protein (hypothetical protein TpM_02g00758_549–577_), some of which were present in multiple samples despite the disparity in the BoLA-I molecules expressed by the three cell lines, was suggestive that a substantial proportion of the longer peptides were coprecipitants rather than peptides eluted from the peptide-binding groove of the purified BoLA-I molecules. 

To confirm the capacity of the *T. parva* peptides to bind to BoLA-I molecules in vitro, binding assays were performed on a subset (*n* = 19) of peptides (Table 2, Figure 1E). This included representative peptides from across a range of NetMHCpan4.1 predictive scores (0.01–22.8%) and at least one allele from each MHC haplotype. For 12/13 of the peptides predicted to be BoLA-I binders, the binding assay confirmed binding; the exception was NSFVTDTFEKL, which had the poorest ranking score of the predicted binders (rank 1.68%, 3*00201). Conversely, three peptides with a rank of >2% bound to the relevant MHCI alleles in vitro, RLFNFATKRI (rank 2.32%), and SLKSALIDTLI (rank 2.47%) bound to 6*01301 and YGDYGEFDRKTK (rank 13.4%) bound to 2*01201; however, all three exhibited weaker binding than peptides that were predicted binders. The three peptides with the poorest rank scores (all rank >14%) failed to exhibit any binding on the assay. A notable feature of the results was the high level of correlation between the predicted percent rank binding scores and the quantitative results observed in the in vitro assay, with the data from the in vitro analysis, therefore, corroborating the BoLA-I binding predictions from NetMHCpan4.1. Interestingly, the in vitro binding assays demonstrated that three peptides identified as 2*01201-binders from the BoLA-A10 sample (1011TP) had the capacity to bind to 6*01301 (A18); however, the level of binding was very low, being generally >10 fold lower than the weakest predicted 6*01301 binding peptide. This may reflect the similarity of the peptide binding motifs of the 2*01201 and 6*01301 alleles [38].

ELISPOT assays using established in vitro autologous *T. parva*-specific CD8^+^ T-cell lines were conducted for the same subset of peptides that had been validated by spectral matching. These CD8^+^ T-cell lines produced potent responses against the positive control immunodominant epitopes (Tp2: TpMuguga_01g00056_49–59_; KSSHGMGKVGK, Tp9: TpMuguga_02g00895_67–75_: AKFPGMKKS, and Tp1: TpMuguga_03g00849_214–224_: VGYPKVKEEML for BoLA-A10, A14, and A18, respectively). Responses were also detected against one of the peptides identified from the pMHCI-elution dataset, TpMuguga_02g00895_67–81_, which contained the previously identified Tp9 epitope recognised by BoLA-A14^+^ animals (data not shown), however, no responses were elicited by any of the other peptides tested.

Thus, from this primary set of samples, a total of 68 *T. parva* peptides were identified, of which 23 were predicted to be BoLA-I binders. Data from spectral matching, BoLA-I binding prediction, and in vitro binding assays confirmed the identity of the pMHCI-eluted peptides and their capacity to bind to the relevant BoLA-I molecules; however, ELISPOT data indicated that only 1/33 of the assayed peptides were recognised by *T. parva*-specific CD8^+^ T-cells derived from an ITM-immunised donor.

### 3.2. Immunopeptidome Analysis of Additional T. parva-Infected Cell Lines

Having verified the capacity of LC-MS analysis of BoLA-I-eluted peptides to identify BoLA-I-presented *T. parva-*derived peptides, a second sample set comprising 10 TP cell lines was studied. This included four cell lines, which between them expressed four additional BoLA-I haplotypes: 2824TP (A19: 2:01601, 6:01402), 5350TP (A20: 2:02601, 3:02701), 2408TP (A15: 1:00901, 2:02501, 4:02401), and 2123TP (A12/A15: −2:00801, 1:01901 1:00901, 2:02501, 4:02401); independent replicate samples derived from cell lines 641TP, 2824TP, 2123TP, 5350TP (denoted by the suffix ‘_rpt’), and duplicate samples from an additional BoLA-A10 cell line (5072TP).

The data generated from the second sample set had a similar profile to that obtained from the preliminary set of samples. A range of between 5333 and 12119 total peptides (average = 8487) and 6–107 *T. parva* peptides (average = 46) were identified in each sample (Figure 2A); thus, the average percentage of *T. parva* peptides in the data was 0.53% (a summary of all of the data is provided in Appendix A). The total number of *T. parva* peptides identified and number of unique sequences identified in all 10 samples combined were 456 and 294, respectively. Details of all *T. parva* peptides identified in the second sample set are provided in Appendix A. As in the preliminary dataset, the *T. parva* peptides exhibited an anomalous length distribution, with only 53% being 8–12 mers and 44.9% being ≥13 mers (1.8% of *T. parva* peptides were 7 mers, *n* = 8), whilst >90% of the bovine peptides were of the canonical 8–12 a.a. length, suggesting the anomalous profile was parasite-specific (Figure 2B). Similarly, the percent rank binding prediction results were similar to the first dataset, with a high proportion of bovine derived 8–12-mer peptides predicted to be binders (for all samples combined = 91%, range in samples = 86–96%), whilst only 57% (range in samples = 20–100%) of the *T. parva*-derived 8–12-mer peptides were predicted to be BoLA-I binders (Figure 2C).

Together, the data indicate that BoLA-I immunopeptidomic analysis of *Theileria parva*-infected cell lines generated a consistent data profile that comprised subsets of peptides of canonical length, of which ~50% were predicted to be MHCI-binders, and peptides of anomalous lengths that contained few MHCI-binders.

### 3.3. Exclusion of Putative Coprecipitating Parasite Proteins and Application of Immunoinformatics Provides a Refined List of Putative BoLA-I-eluted T. parva Peptides

Examination of the collated data derived from the 13 samples demonstrated that ribosomal proteins, histones, and TpM_02g00758 were dominant sources for the 524 *T. parva*-peptides identified, accounting for 27.1%, 19.3%, and 16.2% of the peptide repertoire, respectively. The majority of peptides derived from TpM_02g00758 were of an anomalous length (average length = 17.5 a.a., with 67/85 of the peptides being ≥13 a.a. in length), were predominantly overlapping peptides originating from a small 30 a.a. region (79/85 peptides (92.9%) derived from TpM_02g00758_547–577_), and had a poor percent rank binding prediction score (median = 93.1%, only one peptide had a predicted percent ranking binding score below the 2% threshold). 

Peptides from other proteins exhibited similar, but less pronounced, characteristics—for example, ribosomal protein S28-B 40s (TpMuguga_03g00428_1–14_) and histone H2A variant 1 (TpMuguga_02g00611), as shown in Figure 3. The recurrence of peptides from localised regions of a small subset of proteins in multiple samples of nonsimilar BoLA-I haplotypes, which generally exhibited poor percent rank binding scores and anomalous peptide lengths, supports the designation of these peptides as co-precipitants rather than genuine MHC-binders.

Based on this, the dataset was refined by removing overlapping peptides identified in multiple samples expressing disparate BoLA-I haplotypes (for this purpose, BoLA-A14 and BoLA-A15, which express common BoLA alleles, were grouped together) where one or more of the peptides was not predicted to be a BoLA-I binder (defined as a rank-predicted binding score >20%). When applied to the combined dataset, this refining process removed 55.2% of the *T. parva* peptides (*n* = 289/524—Appendix A). The removed peptides had an average length of 14.7 amino acids and a median rank prediction score of 66.3%, with only 20 (6%) having a rank prediction score of <2%. In contrast, the remaining 235 peptides had an average length of 11.6 amino acids, a median rank prediction score of 1.43%, and the number of peptides with a percent rank prediction score of <2% was 128 (55.4%)—thus, the removal of the coprecipitants had a profound effect on the dataset, leading to a substantial improvement of the predicted percent rank score and making the enrichment for genuine BoLA-I binders in the dataset evident (Figure 4).

These putative coprecipitant peptides were derived from 25 proteins. These included TpM_02g00758, 12 ribosomal proteins, and 3 histones, which together were the source of 88.9% of the coprecipitant peptides (*n* = 257/289). Based on the high representation of these proteins in the coprecipitation pool, it was decided to remove all peptides derived from these classes of proteins. This resulted in the removal of an additional 75 peptides, so that a total of 364 peptides were excluded; this peptide set had an average length of 14 a.a., a median predicted-rank score of 22.8%, and 68 peptides (18.7%) of the peptides had a rank prediction score of <2%. The retained peptide dataset consisted of 160 peptides, with an average peptide length of 11.6 a.a., a median rank predicted binding score of 2.03%, and 80 peptides (50%) that were predicted to be binders. Thus, the removal of all peptides derived from ribosomes, histones, and TpM_02g00758 caused a slight deterioration in the statistics of the retained peptide set but was considered a good compromise to decrease the retention of possible coprecipitant artefacts. As a final step to refine the peptide dataset, an immunoinformatics filter was used, and all remaining peptides that had a predicted percent rank binding score of >2% (i.e., not predicted MHCI-binders) were removed. This left a final dataset of 80 peptides, which, after consolidation of overlapping, nesting, and duplicate identifications, resulted in 74 unique peptides from 68 proteins (Table 3).

### 3.4. Analysis of the Reproducibility of the Identified T. parva BoLA-I Immunopeptidomes

In the final dataset, the average number of *T. parva* peptides identified per sample was approximately six, suggesting that only a small subset of the BoLA-I-presented *T. parva*-peptides had been identified. To evaluate what effect this had on the reproducibility of the *T. parva* peptidomes described, we examined the overlap of peptides identified in cell lines that had been subjected to duplicate analysis of independent samples (technical duplicates for 641TP, 2824TP, and 5350TP, respectively) and in the triplicate datasets from TP cell lines expressing BoLA-A10 and BoLA-A15 haplotypes (comprising the 1011TP/5072TP and 2123TP/2408TP samples, respectively).

For the BoLA-A18, A10, and A15 groups, there was partial, but limited, overlap between replicate samples; in contrast, for the BoLA-A19 and BoLA-A20 groups, there was no overlap between the samples (Figure 5). As a summary statistic, the percentage of *T. parva* peptides identified in replicate samples was 7.3%; in comparison, the overlap between the bovine peptidomes from the same samples was greater, with 47.1% of bovine peptides identified in replicate samples. The low level of overlap observed between replicate *T. parva* immunopeptidomes is most likely a consequence of the low number of *T. parva* peptides identified (notably in the BoLA-A19 and BoLA-A20 groups, only one peptide was identified in one of the replicate samples); however, the identification of a subset of *T. parva* peptides in replicate samples indicates that the immunopeptidomes described in this study are at least partially reproducible, and higher resolution studies, yielding greater depth of peptide repertoire coverage, would likely produce datasets exhibiting greater reproducibility.

### 3.5. Analysis of T. parva Peptides Presented by BoLA-DR

We sought to expand the immunopeptidiomic analysis to bovine MHCII molecules. Cattle express two BoLA-II isotypes—DR and DQ. The peptide-binding groove of MHCII molecules is formed by a combination of the coexpressed α and β chains that form the MHCII heterodimer. Both BoLA-DQA and DQB loci exhibit polymorphism and are duplicated in some BoLA-haplotypes [39], whereas the BoLA-DRA locus is monomorphic and there is only a single function and expressed BoLA-DRB locus [40,41]. Consequently, immunopeptidomic analysis of BoLA-DR was considered less complex, and we undertook an analysis of the peptides eluted from BoLA-DR molecules of six *T*. *parva*-infected cell lines: 2123TP (BoLA-DR 15:01/11:01), 2824TP (BoLA-DR 16:01), 5072TP (BoLA-DR 10:01), 641TP (BoLA-DR 20:02), 495TP (BoLA-DR 10:01/11:01), and 5350TP (BoLA-DR 12:01), which between them expressed six different BoLA-DR molecules (Appendix A). The total number of peptides identified in each sample ranged from 4592 to 8547 peptides (average = 6738, Figure 6A). After filtering sequences with close homology to the bovine proteome, a range of 58–151 peptides (average = 101) were identified as being unambiguously derived from *T. parva*, representing 1.5% of the total peptides identified (total number and number of unique *T. parva* peptides identified were 607 and 326, respectively; Appendix A). The average length of *T. parva* peptides was slightly shorter (15.0 a.a.) than the bovine peptides (15.7 a.a) (Figure 6B) and adhered less to a classic Gaussian distribution. The proportion of peptides that were predicted to be binders (i.e., had a percent rank predicted binding score of <5% when using NetBoLAIIpan; the threshold used for BoLA-DR binding) for 13–21-mer bovine peptides were consistently high, ranging from 82 to 84% (average = 83%). In contrast, the proportion of *T. parva*-derived 13–21-mer peptides that were predicted to be BoLA-DR binders was much lower, ranging from 5 to 22% (average = 12%; Figure 6C).

Similar to the BoLA-I data, a notable feature of the *T. parva* peptides in the BoLA-DR dataset was the dominant representation of peptides from a small subset of proteins. This included TpMuguga_02g00758, from which peptides were identified in all six samples and which accounted for a total of 118 peptides (19.4% of all *T. parva* peptides in the BoLA-DR dataset). As with the BoLA-I data, the peptides from these proteins identified in different samples were often clustered in specific regions, were overlapping, and predominantly had poor predicted percent rank binding scores, indicative of coprecipitating peptides. Application of the same process as described for the BoLA-I data to identify putative coprecipitants suggested that a substantial majority of the *T. parva* peptides (82.9%, *n* = 503/608—Appendix A) were coprecipitants. These peptides were derived from 36 individual proteins, of which 25 were either ribosomal or histone proteins. A comparison of those proteins that were identified as the sources of coprecipitant peptides in the BoLA-I and BoLA-DR datasets showed a high level of convergence (15 proteins common to both) and a correlation in the number of peptides that individual proteins contributed to the BoLA-I and BoLA-DR datasets (Figure 7A and Appendix A). 

The removal of coprecipitated peptides had a limited impact on the average predicted rank percent binding score of the combined BoLA-DR peptide dataset (46.3% vs. 46.9%), however, the distribution of the retained peptides showed a clear bimodal pattern with peaks of peptides with a rank percent prediction score of <5% and >95%; in contrast, the profile of the coprecipitated *T. parva* peptides showed no evidence of selection of predicted BoLA-DR binders, with only a dominant peak for peptides with a predicted rank percent binding score of >95%—Figure 7B. Although removal of the coprecipitants provided an enhanced dataset, the majority of the peptides retained in the dataset were not predicted to be BoLA-DR binders; 79/104 (76%) of the peptides had a predicted rank binding score of >5%. As with the BoLA-I data, all peptides derived from TpM_02g00758, ribosomal proteins, and histones were removed (34 peptides with a median rank predicted binding score of 46.95, and only two peptides were predicted to be BoLA-DR binders), and the default threshold (i.e., a percent rank-predicted binding score of <5%) used to predict MHC binding was applied to generate a final list of BoLA-DR presented *T. parva* peptides. After consolidation of the nested peptides, this list included 15 peptides, each derived from a different *T. parva* protein (Table 4).

### 3.6. Comparison of T. parva Peptidome Data with Previously Identified T. parva Antigens

Recent work applying conventional antigen-screening techniques to identify CD4^+^ and CD8^+^ T-cell epitopes with a peptide library covering 502 *T. parva* proteins from the reference Muguga strain [13] has expanded the number of known T-cell antigens to 36; twenty CD4^+^ T-cell antigens, 10 CD8^+^ T-cell antigens, and six antigens containing epitopes for both CD4^+^ and CD8^+^ T-cells [12,13]. The library included peptides covering 19 out of the 105 proteins that were identified as sources of BoLA-I and/or BoLA-DR presented peptides in this study. Although none of the peptides from these 19 proteins matched experimentally mapped CD8^+^ or CD4^+^ T-cell epitopes, five of the proteins have been identified as T-cell antigens (Table 5). This includes TpMuguga_02g00123 (Tp32—DEAD/DEAH box helicase) and TpMuguga_02g00895 (Tp9), which have been shown to contain epitopes for both CD8^+^ and CD4^+^ T-cells. Thus, 26.3% of the proteins identified as sources of MHCI/MHCII presented peptides from immunopeptidomics that have been included in the recent conventional antigen-screening study [13] have been demonstrated to contain recognised epitopes. In contrast, only 7.5% of the proteins selected for inclusion in that study were shown to contain epitopes. This suggests that, although only one of the peptides identified in this study has been validated as containing an epitope, immunopeptidomics could be used to preferentially select proteins that are sources of CD4^+^/CD8^+^ T cell antigens. 

## 4. Discussion

Identification of T-cell antigens for inclusion in vaccines against complex pathogens remains challenging. The potential for immunopeptidomics to address this critical obstacle in the development of novel subunit vaccines against a wide range of pathogens, especially nonviral pathogens with complex proteomes [42], is receiving greater attention. This is of especial relevance to intracellular eukaryotic pathogens, such as *Plasmodium,* where integration of data from immunopeptidomics with data from other antigen-identification approaches has been advocated as the most efficient way in which to identify antigens with potential for vaccination for pre-erythrocytic malaria vaccines [43].

Antigen-identification persists as a constraint on the development of novel *T. parva* subunit vaccines with the capacity to induce T-cell responses. Conventional screening approaches have yielded a number of *T. parva* antigens for both CD4^+^ and CD8^+^ T-cells [12,13,44] on some MHC backgrounds, and in a recent study, use of a purely ‘immuno-informatic’ approach to identifying T-cell antigens from *Theileria* has been attempted [45]. In this study, we provide the first description of MHC-peptide elution studies being used to investigate the immunopeptidome of *T. parva*-infected cells to define peptides presented by BoLA-I and BoLA-DR molecules. 

Although the large proteome and high parasite diversity are disadvantages in conventional antigen-screening approaches, the biology of *T. parva* has inherent advantages with regards to the application of immunopeptidomics. Firstly, it is easy to generate and maintain rapidly and indefinitely proliferating *T. parva*-infected cells in vitro—enabling the accumulation of sufficient cells (>1 × 10^9^ infected cells), with an infection rate of ~100%, from the target host species. In malaria, the application of immunopeptidomics has been hindered by the low level of hepatocyte infection (<10%) that can be achieved [43]. The second feature of *T. parva* is that the in vitro infected cells express high levels of both BoLA-I and BoLA-II and are of the same phenotype as the cells infected in vivo (i.e., T and B lymphocytes). The capacity of these *T. parva*-infected cells to recall antigen-specific T-cells from recovered animals without the need for supplementary APCs confirms that the peptides presented in the context of BoLA-I/BoLA-II by *T. parva*-infected cells are those that stimulate T-cell responses [3,29]. As such, cells infected in vitro with *T. parva* are highly appropriate for studies to apply immunopeptidomics and avoid the potential complication of cell types selected for analysis that was observed in recent *Chlamydia* studies, where dendritic and epithelial cells were found to have discordant and nonoverlapping immunopeptidomes [46]. 

In this study, we exploited the features of *T. parva* to analyse the immunopeptidomes from a total of nine different cell lines representing a range of BoLA-I haplotypes and BoLA-DR molecules. This gave our study a unique structure compared to immunopeptidomic studies that have been reported for other eukaryotic parasites (*Leishmania*, *Plasmodium*, and *Toxoplasma*), where data have been generated for either MHCI or MHCII and from only single samples [19,20,21]. This approach was primarily driven by two factors. Firstly, as cattle are not a ‘model’ species, there was, at the outset of these studies, very limited information on the peptide binding motifs of bovine MHC molecules—the inclusion of multiple BoLA-I/BoLA-DR genotypes allowed us to define their peptide-binding motifs [32,38,47], which will find applications in enhancing ‘immuno-informatic’ based studies in cattle (and was also pivotal in subsequently refining the data presented herein). Secondly, the ultimate aim of the study was the exploration of the feasibility of using immunopeptidome analysis as an alternative and/or complementary approach contributing to CD4^+^/CD8^+^ T-cell candidate antigen identification for a *T. parva* vaccine that will be used in outbred cattle populations. As such, there was an interest in generating data for both BoLA-I and BoLA-II molecules for a range of different genotypes. In this study, we purposely focused on MHC haplotypes present in Holstein-Friesian cattle. This was primarily because these high-yielding dairy cattle and their crosses with indigenous cattle continue to increase in numbers in regions of Africa where ECF occurs. These high-value animals are considered critical to improving agricultural production and meeting the growing local demand for dairy products. As an ‘exotic’ breed that shows minimal tolerance to *T. parva* infections, these animals are particularly susceptible to ECF and are, therefore, the primary target for any vaccine against *T. parva*. Secondly, most previous antigen-identification work has primarily focused on animals expressing MHC haplotypes present in Holstein-Friesian animals. Thirdly, at the moment there is insufficient data on the diversity and frequency of MHC genotypes in cattle breeds indigenous to *T. parva-*endemic regions on which to base the selection of samples to subject to immunopeptidomic analysis in these populations. This fundamental gap in our knowledge is currently being addressed by several groups through the development and application of high-throughput sequencing approaches for exploring MHC diversity in a range of cattle populations [28,48,49].

A consistent feature of the BoLA-I and BoLA-DR *T. parva*-derived peptide datasets obtained during this study was the predominance of peptides that were of noncanonical length and/or predicted to be non-MHC binders. Equivalent anomalous peptides were not a feature of the bovine peptidomes, suggesting that these anomalous peptides were specific to the pathogen rather than a technical fault in the protocol. Non-genuine MHC binders that co-precipitate with pMHC molecules in elution studies using equivalent protocols have been documented before (e.g., in HIV-1 [50]). However, the extent to which putative co-precipitants dominated the *T. parva* datasets was remarkable; a total of 55.2% and 82.9% of the peptides identified from the BoLA-I and BoLA-DR datasets were defined as co-precipitants in this study. The co-precipitant peptides came from a small subset of proteins, which showed a high level of overlap between the BoLA-I and BoLA-DR datasets, suggesting that co-precipitating peptides did not represent random ‘noise’ but rather a feature of specific *T. parva* proteins. TpMuguga_02g00758, a hypothetical protein of 640 amino acids, was the largest contributor of co-precipitating peptides to both datasets (29.5% and 23.3% in the BoLA-I and BoLA-DR datasets, respectively). The second and third most common proteins acting as sources of co-precipitants in both datasets were histones (contributing a total of 28.5% and 24.5% of co-precipitants in MHCI and DR, respectively), whilst the most commonly represented family of proteins were ribosomal proteins (11 and 21 proteins contributing 25% and 39.4% of coprecipitating peptides in the BoLA-I and BoLA-DR datasets, respectively). Together, TpMuguga_02g00758, histones, and ribosomes constituted >80% of the co-precipitants identified in both datasets. 

Relatively few immunopeptidomic studies have been conducted on parasite-infected cells [19,20,21], so there is little comparative data to assess whether this magnitude of coprecipitating peptides is a feature common to intra-cellular parasites or specific to *Theileria* (only eukaryotes express both ribosomes and histones, preventing direct comparison with data from viral or bacterial studies). Coprecipitants were not specifically examined in any of the other parasite immunopeptidomic studies. However, direct comparison between the studies would perhaps be confounded by technical differences; in the *Toxoplasma gondii* study [20], a monoallelic secreted MHCI model was used and in the *Plasmodium* study [21], DCs were loaded by incubation with parasitized RBCs, which may, through different mechanisms, change the quantity and/or source of any coprecipitants observed. The high level of coprecipitants observed for *T. parva* may reflect fundamental biological differences between it and the other parasites for which immunopeptidomic studies have been completed. Foremost of these is the location of *T. parva* schizonts free within the host cell cytoplasm (*Plasmodium*, *Toxoplasma*, and *Leishmania* reside within vacuoles). This feature, combined with the active secretion of *Theileria* proteins into the host cell cytoplasm [51,52,53] and the close interaction between the parasite and various components of the host cell’s architecture (e.g., attachment to the host cell’s mitotic spindle during cell division [54] and the formation of various host-pathogen complexes on the schizont surface [53], may afford greater opportunities for *T. parva* proteins to non-specifically bind to MHC molecules and/or integrate into elements of the host cell’s structures that are subsequently co-precipitated during the elution protocol. Notably, the TpMuguga_02g00758 gene product has a signal peptide, indicating that it is likely to be secreted into the host-cell cytoplasm. However, absence of any other annotated features of this protein precludes any inferences about its biological function and how this might relate to it being the major co-precipitant in both the BoLA-I and BoLA-DR datasets. A second pertinent feature of *T. parva* biology is the transformation of the host cell and rapid proliferation of both the host cell and parasite. To sustain the DNA and protein production that proliferation requires, histones and ribosomes are likely to be subject to high levels of expression and turn-over which may contribute to their overrepresentation as coprecipitants. At present there is no proteome of the schizont stage of *T. parva*, however transcriptomic analyses of *T. parva* schizonts show that several histone and ribosomal proteins are amongst the most abundantly expressed proteins [55]. Comparison of these data with proteomic analysis of schizonts of the closely related *T. annulata* has shown a high level of concordance—suggesting the transcriptome is quantitatively representative of the *T. parva* schizont proteome [56].

After removal of co-precipitants, both the BoLA-I and BoLA-DR datasets showed bias for peptides that were predicted to be strong MHC-binders (Figure 4 and Figure 7B), demonstrating preferential selection of MHC-binding peptides. Within these datasets there remained a high number of peptides derived from TpMuguga_02g00758, ribosomes and histones. Based on the high frequency at which these proteins were the source of co-precipitating peptides we decided to exclude all peptides from these proteins. Whilst this did not enhance the parameters by which the quality of the datasets were measured, it removed classes of proteins that were evidently sources of co-precipitating proteins and so compromised as potential candidate vaccine antigens. Following removal of these peptides, the BoLA-I and -DR datasets were reduced substantially from 524 and 607 to 160 and 70 respectively; however, within these there remained a substantial proportion of peptides that were predicted to be non-MHC binders. It is possible that these peptides may also be co-precipitants; however, there was no clear rationale in the current dataset for defining them as such and so they were excluded from the final peptide selections solely on the basis of default thresholds of predicted rank percent scores used to define BoLA-I (<2%) and BoLA-DR (<5%) binding. The high correlation seen between the predicted % rank binding scores and the results from the in vitro binding assay (Table 2), the very high proportion of bovine derived ligands with low rank scores (indicating BoLA binding) [38] and the observation that most known CD4^+^ and CD8^+^ T-cell epitopes from *T. parva* proteins have predicted % rank binding scores within the respective 5% and 2% thresholds [32,38], support this approach to rationalising the final peptide lists. The availability of well-defined peptide-binding motifs for these bovine MHC molecules was critical in curation of the datasets and attests to the importance of high quality immuno-informatic data to support future immunopeptidomic studies.

Recognition by T-cells from immune animals has been used as a standard component of assays attempting to ‘validate’ peptides identified from immunopeptidomic studies. On assaying a subset of the *T. parva* peptides identified from the first 3 BoLA-I samples only 1 out of 33 of the peptides (3.3%) was recognised. *Prima facie*, this lack of recognition was disappointing. However, it is similar to results from other immunopeptidomic studies (e.g., [57,58]) and in the context of immunodominance, which is a characteristic feature of many CD8+ T-cell responses [59], is to be anticipated. Due to factors such as TCR repertoire and APC competition [60], although a broad range of peptides may be presented by MHC molecules (and so reported in the immunopeptidome), immunodominance causes a detectable T-cell response against only a small subset of the presented peptides. Previous work by our group has confirmed that *T. parva*-specific CD8^+^ T-cell responses are subject to immunodominance, with up to 78% of the responses in BoLA-A10 and A18 homozygous animals being directed against single peptides [15]. The failure of the peptides identified in the immunopeptidome to be recognised by T-cells from immune animals does not discredit them as vaccine candidates. Studies in a range of pathogens including *Trypanosoma cruzi* [61], *Mycobacterium tuberculosis* [62,63] and various viruses [64,65,66] has shown that subdominant/cryptic epitopes can confer protection, which in some cases is better than that afforded by immunodominant epitopes. Furthermore, it has been shown in numerous studies [61,67,68,69,70,71] across a similar range of pathogens, that induction of T-cell responses in the absence of immunodominant antigens (as can be engineered in subunit vaccines) can successfully elicit responses against epitopes that are normally ‘cryptic’/subdominant following natural infection (which, in the context of *T. parva*, ITM immunisation essentially is). Thus, with regards to using immunopeptidomics to identify novel vaccine candidate antigens, the concept of using recognition of peptides by T-cells from immune animals to ‘validate’ peptides is conceptually flawed. To discount peptides not recognised by T-cells of immune cattle would mitigate against a major benefit of immunopeptidomics, the capacity to identify a large repertoire of MHC-presented peptides, amongst which may be potential epitopes that cannot be identified from T-cell screening but which potentially have the capacity to confer protection when utilised in vaccines. Data from recent studies of human melanoma [57] and African Swine Fever Virus [58] have confirmed T-cells recognising peptides identified through immunopeptidomics but which were ‘cryptic’ (i.e., not recognised by T-cells from the affected individuals/infected pigs) can successfully be induced by vaccination with these peptides. In future studies we plan to generate a multi-epitope vaccine construct including a selection of the peptides identified herein to directly assess their potential as vaccine candidates (see below). Encouragingly, evidence from a recent, unrelated study implies that it will be possible to induce T-cell responses against *T. parva* epitopes that are not recognised following natural infection. Following Adenovirus/MVA heterologous prime-boost immunisation of 4 calves with *T. parva* antigens (Tp2, Tp9 and Tp10) all 4 animals expressing either BoLA-DRB3*11:01 and/or 10:01 generated CD4^+^ T-cell responses specific for the Tp10 (TpMuguga_04g00772) antigen, which had not been previously observed as a CD4^+^ T-cell antigen in ITM-immunised animals bearing these or other BoLA-DR alleles [13]. These Tp10-specific CD4^+^ T-cells were capable of recognising *T. parva*-infected cells and, as far as could be determined, were functionally similar to CD4^+^ T-cells against immunodominant epitopes induced by ITM. 

The *T. parva* cell lines chosen for the initial immunopeptidomic analysis were selected partly because immunodominant CD8^+^ T-cell epitopes restricted by MHCI proteins expressed by these cells (BoLA-A10+, BoLA-A18+ and BoLA-A14+) had already been defined. However, only the BoLA-A14-restricted TpMuguga_02g00895_67–75_ epitope was identified in the immunopeptidome (as a nested peptide within a 15-mer). This most likely was a reflection that the immunopeptidomes characterised in this study were only partial. In the final dataset the average number of *T. parva* unique peptides identified per BoLA-I and BoLA-DR sample was only ~6 and ~2.5 respectively. Examination of replicate samples demonstrated very limited overlap in the BoLA-I immunopeptidomes described (Figure 5—only 7.3% of the peptides were identified in multiple samples)—confirming that only a fraction of the *T. parva* peptidome was being identified from any individual sample at the depth of data generated in this study. A more intensive immunopeptidome analysis would be expected to enable a more complete characterisation of *T. parva* immunopeptidomes, which would potentially include the identification of the known *T. parva* CD4^+^ and CD8^+^ epitopes. For reasons given above, we elected in this study to prioritise the inclusion of multiple BoLA genotypes. However, future work complementing this with ‘deep’ immunopeptidomic profiling for a limited number of samples would be beneficial.

The ultimate aim of the study was to assess the feasibility of immunopeptidomics to identify candidate antigens to include in novel subunit vaccines. At the end of the analysis, a total of 74 and 15 unique peptides with predicted high binding capacities for BoLA-I and BoLA-DR molecules were identified. This information can be used in multiple ways to inform candidate antigen selection. For example, as described in the final results section, immunopeptidomic data could increase the efficiency of conventional antigen screening by focusing on *T. parva* proteins that are evidently contributing to the immunopeptidome. This may be of particular value as other parameters that have been evaluated (e.g., transcript abundance, presence of signal peptides) have not proved reliable for targeting antigen screening [13]. The simplest and most direct application of the data would be to include the identified peptides in a vaccine delivery platform (e.g., an adenoviral-vectored vaccine) and administer this to animals of the appropriate BoLA genotypes to assess immunogenicity and/or protective efficacy. However, this approach would only partially utilise the information made available from the immunopeptidomic data. The well-documented antigenic diversity of *T. parva* [8,9,10,11] and the polymorphism of bovine MHC in cattle in *T. parva*-endemic areas (Vasoya et al. submitted) preclude the direct application of immunopeptidomic studies to identify specific epitopes for all the potential permutations of MHC genotype and parasite strain. A more practicable approach would be to integrate this and future immunopeptidome data identifying proteins preferentially accessing the MHC processing pathways with data analysing the host immunogenetic and pathogen genetic diversity. The repertoire of bovine BoLA genotypes in cattle populations in *T. parva*-endemic areas is now being analysed using high-throughput sequencing approaches (Vasoya et al. submitted), and work has begun to develop improved immuno-informatics algorithms for cattle [32,38,47] enabling accurate prediction of binding motifs of BoLA-I and BoLA-DR molecules. Simultaneously, transcriptomic and genomic sequencing of multiple *T. parva* strains [8,14] has recently been undertaken, enabling the strain diversity of *T. parva* proteins to be assessed. Together, the outputs of these ‘omics’ technologies could be utilised to identify candidate *T. parva* antigens presented on BoLA-I/BoLA-II (immunopeptidomics) and evaluate them for their degree of conservation (genomics/transcriptomics of *T. parva* strains) and their content of peptides (immunoinformatics) that can be presented on multiple BoLA-I and BoLA-DR genotypes (MHC repertoire analyses). The complexity of both the host MHC and pathogen strain diversity will undoubtedly make identification of a comprehensive set of candidate antigens immensely challenging, but this approach will make the best use of the data being generated from a suite of ‘omics technologies and provide a rational approach to identifying a ‘minimal’ set of antigens that would enable the induction of CD8^+^ and CD4^+^ T-cells from animals bearing a range of different MHC genotypes against diverse *T. parva* strains. 

In this study, we have provided the first exploration of the immunopeptidome of *T. parva*-infected cells. The study encompasses data from multiple BoLA-I and BoLA-DR genotypes, identifying 74 and 15 unique BoLA-I and BoLA-DR binding peptides, and as such forms probably the most comprehensive immunopeptidomic analysis of any eukaryotic pathogen to date. However, there is clear evidence that the characterisation of the immunopeptidome completed herein is only partial and further studies are required to gain a more comprehensive understanding of the *T. parva* BoLA-I and BoLA-II immunopeptidomes. Based on the dataset obtained in this study, we have developed a simplified curation system that can be applied in future *T. parva* immunopeptidomic studies to effectively remove coprecipitating peptides that can dominate eluted peptide datasets and restrict the data to peptides that are predicted to be genuine MHC-binders. Due to the high levels of diversity in both *T. parva* and bovine MHC genotypes, we propose that the information derived from immunopeptidomics can most productively be used in the development of novel vaccines, not as a means to identify individual epitopes that are specific to a particular combination of *T. parva* strain and bovine MHC allele, but rather as part of an integrated strategy that aims to identify a repertoire of proteins that can potentially elicit CD4^+^ and CD8^+^ T-cell responses across a breadth of cattle MHC genotypes and that are able to provide protection against the spectrum of *T. parva* strains present in endemic areas. Such an approach will build on the synergy of the multiple new technologies that are being used to study *T. parva* and bovine immunogenetics and offer new opportunities to tackle the conundrum of T-cell antigen identification, which remains an obstacle to developing novel vaccines against this important disease.

## Figures and Tables

**Figure 1 vaccines-10-01907-f001:**
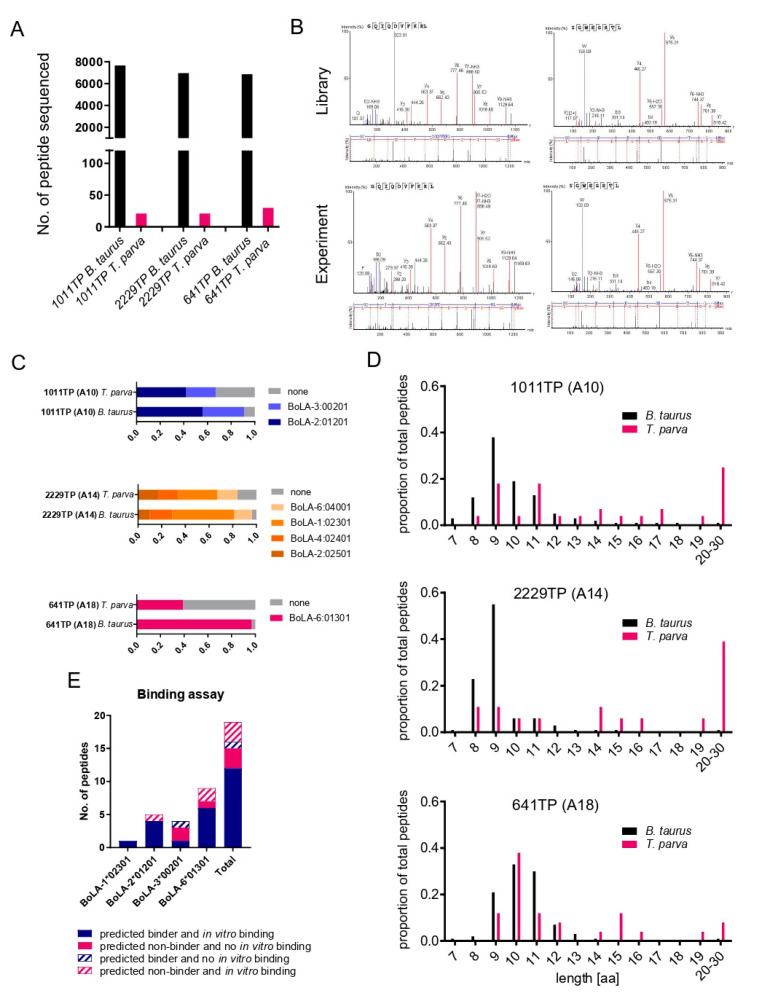
Characteristics of BoLA-I-eluted peptides from 1011TP, 2229TP, and 641TP cell lines. (**A**) Number of *B. taurus* and *T. parva* peptide sequences identified in each sample. (**B**) Spectra obtained from the experiment (bottom), and synthetic library (top), respectively, for two example peptides that were mapped to the *T. parva* proteome. The most abundant, annotated b- (N-terminal) and y- (C-terminal) fragment ions are indicated together with the measured mass over charge ratio. (**C**) Binding predictions of *B. taurus* and *T. parva* 8–12 mer peptides stratified by their predicted BoLA-I allele of origin (peptides with a rank predicted binding score of <2% to a BoLA-I allele were considered to be binders). Peptides predicted to be MHC-binders are represented by coloured blocks, peptides predicted to be non-MHC binders are represented by grey blocks; the size of the blocks is proportional to the number of peptides in the respective datasets. (**D**) Length distributions of *B. taurus* and *T. parva* peptides were identified in each sample. The horizontal axis shows the length of peptides, and the vertical axis shows the proportion of the peptides identified for each length. (**E**) A summary of results from a subset of peptides assayed in an in vitro BoLA-I binding assay is presented in Table 2. For each BoLA allele and for all of the BoLA alleles combined (Total), the number of peptides for which the in vitro assay corroborated the in silico predicted capacity to bind to BoLA-I are shown as filled bars, whilst peptides for which the results from the in vitro binding did not support the in silico prediction are shown as hatched bars, as described in the legend.

**Figure 2 vaccines-10-01907-f002:**
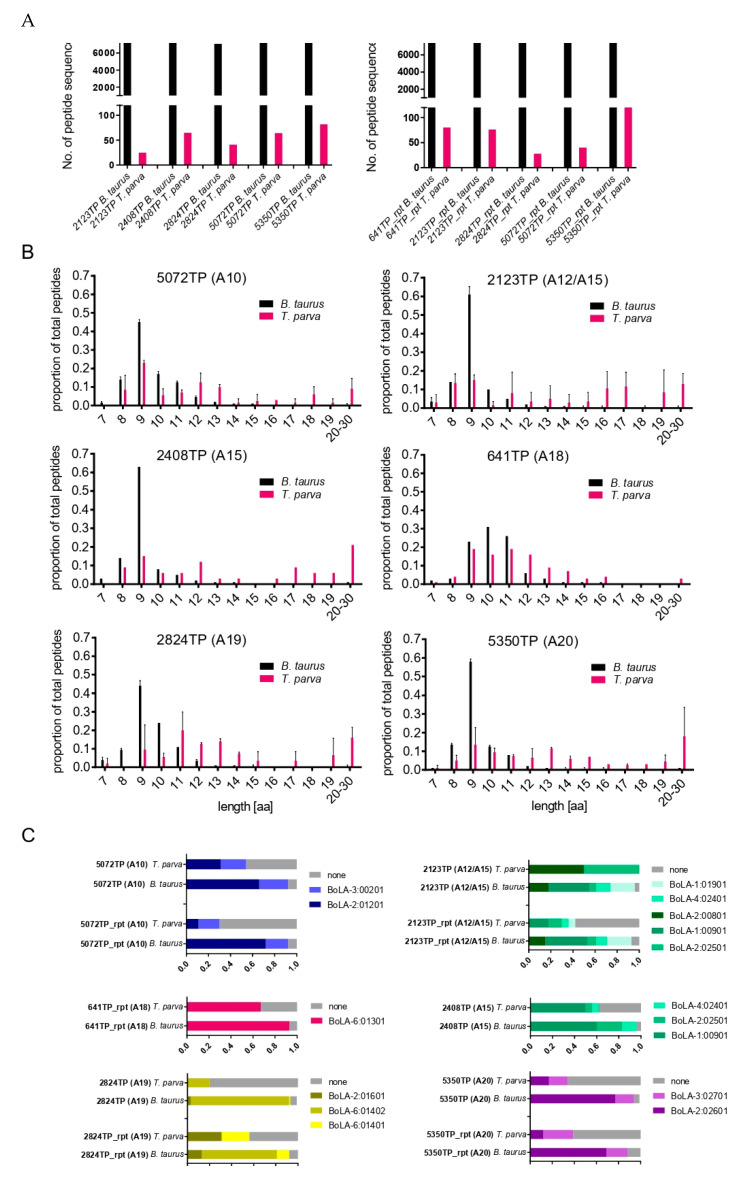
Characteristics of BoLA-I-eluted peptides from 5072TP/5072TP_rpt (A10), 2123TP/2123TP_rpt (A12/A15), 2408TP (A15), 2824TP/2824TP_rpt (A19), 641TP_rpt (A18), and 5350TP/5350TP_rpt (A20) cell lines. (**A**) Number of *B. taurus* and *T. parva* peptide sequences identified in each sample. (**B**) Length distributions of *B. taurus* and *T. parva* peptides identified in each sample. The horizontal axis shows the length of peptides, and the vertical axis shows the proportion of the peptides identified for each length. For samples that had been analysed in two independent elutions, the data has been amalgamated into a single panel. (**C**) Binding predictions of *B. taurus* and *T. parva* 8–12-mer peptides stratified by their predicted BoLA-I allele of origin (peptides with a rank predicted binding score of <2% to a BoLA-I allele were considered to be binders). Peptides predicted to be MHC-binders are represented by coloured blocks, peptides predicted to be non-MHC binders are represented by grey blocks; the size of the blocks is proportional to the number of peptides in the respective datasets.

**Figure 3 vaccines-10-01907-f003:**
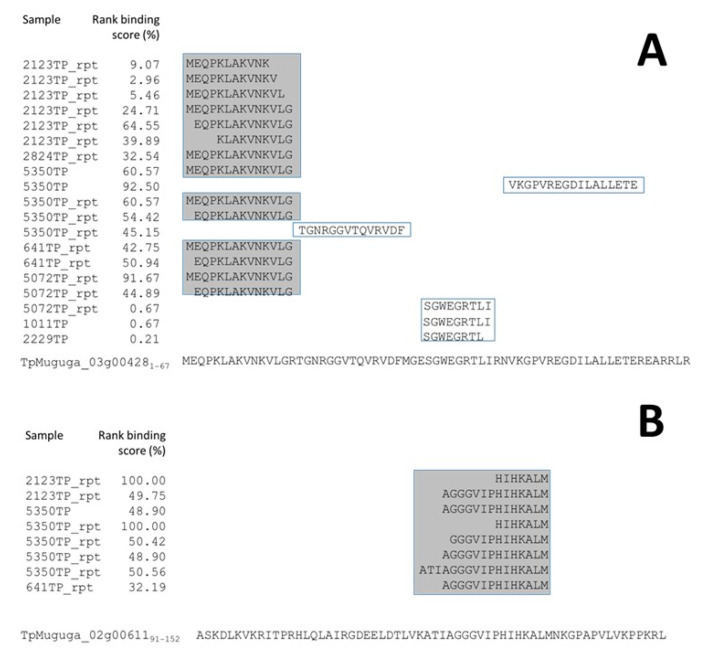
Alignment of peptides identified from (**A**) TpMuguga_03g00428 (ribosomal protein S28-B40S) and (**B**) TpMuguga_02g00611 (histone H2A variant 1) in BoLA-I elution datasets. (**A**) The 19 peptides identified from 8 different samples are shown aligned against the parent protein. Fourteen of the peptides (from 6 different samples representing 4 different MHCI haplotypes) are found as overlapping peptides in the 1–14 region. The median rank-predicted binding score of these peptides is 43.8% (range = 2.96–64.55%), with none predicted to be BoLA-I binders. The mean length of the peptides was 13 a.a. (**B**) The 8 peptides identified from 4 different samples (representing 3 different MHCI haplotypes) are shown aligned against the parent protein. All 8 peptides are overlapping peptides in the 121–137 region. The median rank-predicted score of these peptides is 50.1% (range = 32.2–100%), with none predicted to be BoLA-I binders. None of the peptides were of the 8–12 a.a. length of canonical MHCI-binding peptides.

**Figure 4 vaccines-10-01907-f004:**
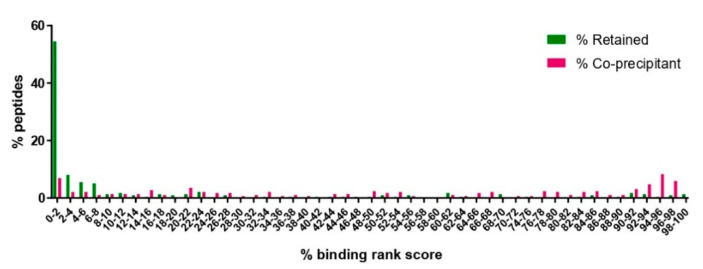
The distribution of predicted percent rank binding scores for peptides considered to be co-precipitants and peptides retained in the combined BoLA-I dataset after removal of the coprecipitating peptides. Peptides with a predicted percent rank binding score of <2% are considered to be binders. A small number of short peptides (<8 amino acids long, *n* = 8) that did not receive a predicted percent rank binding score were ascribed a default value of 100%.

**Figure 5 vaccines-10-01907-f005:**
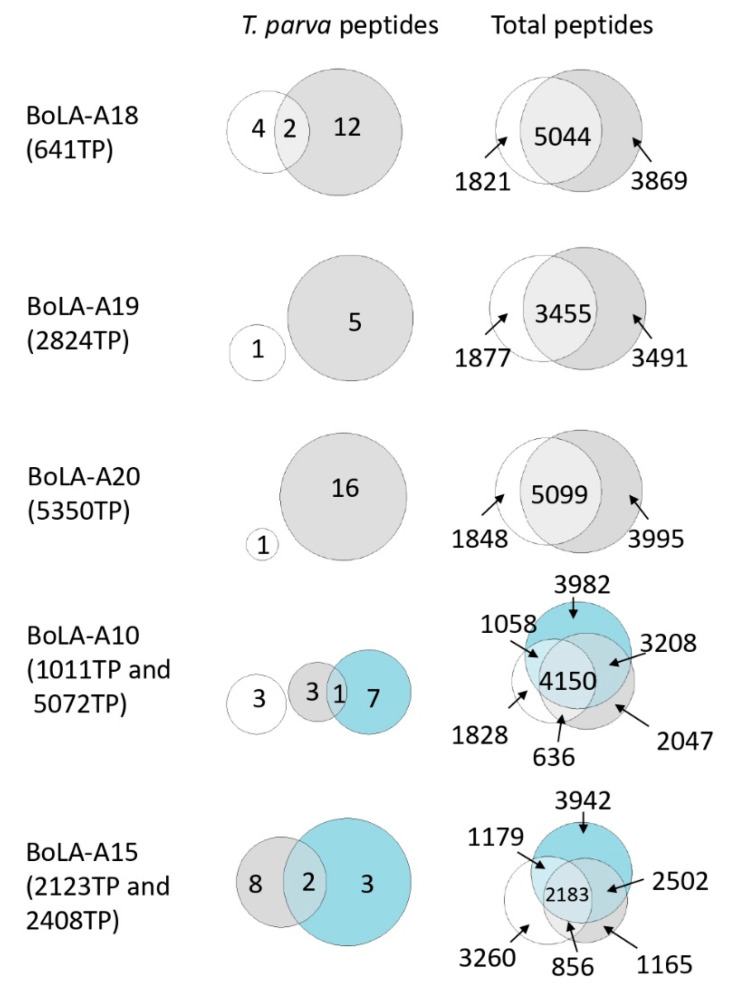
Overlap between the peptidomes identified from duplicate analysis of BoLA-A18, BoLA-A19, and BoLA-A20 samples and from samples sharing the BoLA-A10 and BoLA-A15 haplotypes. Euler diagrams displaying the overlap in the *T. parva* (left) and total (right) peptidomes of duplicated samples or samples sharing BoLA-I haplotypes. The number of peptides that are unique to each sample and shared between samples is indicated. The peptides identified in replicate samples were: BoLA-A18—RMDDKSGGLL from TpMuguga_01g00736 (hypothetical protein) and GEFEKKYIPTL from TpMuguga_01g00757 (Ras family protein); BoLA-A10—AGVELDTQKKFL from TpMuguga_03g00858 (multiprotein-bridging factor 1c); BoLA-A15—FEYEFPINH from TpMuguga_01g00471 (bifunctional thioredoxin reductase/thioredoxin); and EEIAHVLHY from TpMuguga_01g02030 (hypothetical protein). Note that in the BoLA-A15 group, there were no *T. parva* peptides identified in sample 2123TP and so this sample is not represented in the *T. parva* euler diagram.

**Figure 6 vaccines-10-01907-f006:**
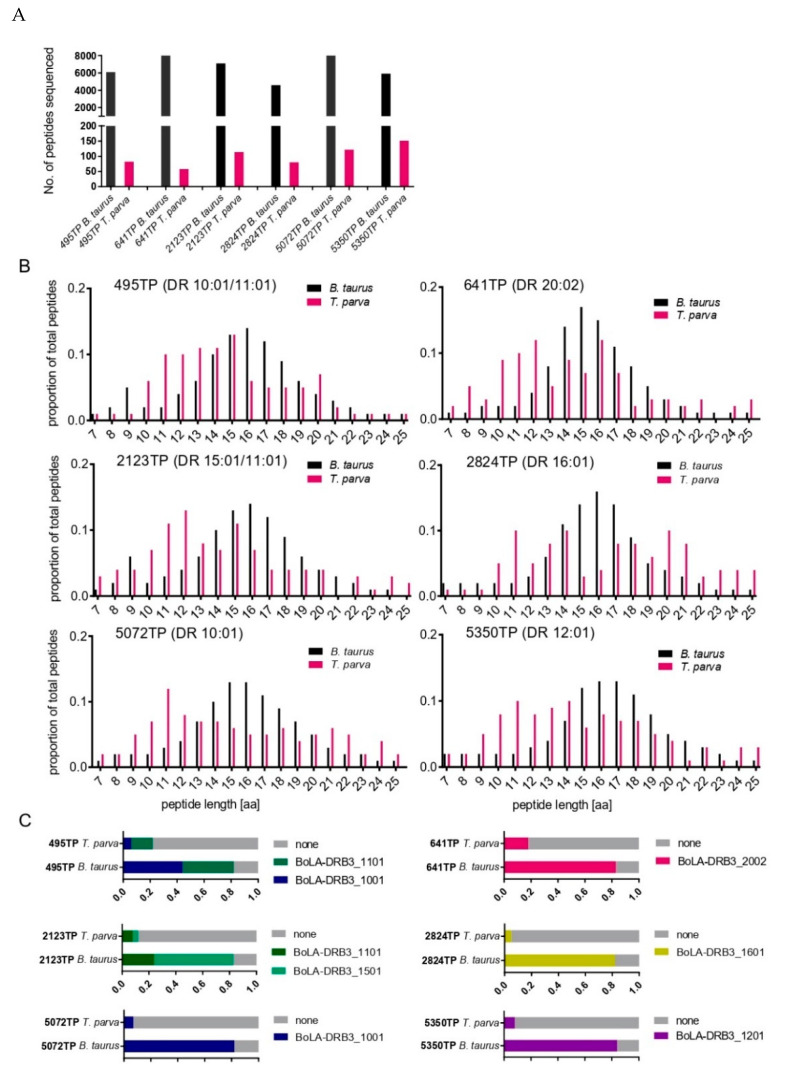
Characteristics of BoLA-DR eluted peptides from 2123TP (BoLA-DR 15:01/11:01), 2824TP (BoLA-DR 16:01), 5072TP (BoLA-DR 10:01), 641TP (BoLA-DR 20:02), 495TP (BoLA-DR 10:01/11:01), and 5350TP (BoLA-DR 12:01) cell lines. (**A**) Number of *B. taurus* and *T. parva* peptide sequences identified in each sample. (**B**) Length distributions of *B. taurus* and *T. parva* peptides identified in each sample. The horizontal axis shows the length of peptides, and the vertical axis shows the proportion of the peptides identified for each length. (**C**) Binding predictions of *B. taurus* and *T. parva* 13–21-mer peptides stratified by their predicted BoLA-DR allele of origin (peptides with a rank-predicted binding score of <5% to a BoLA-DR allele were considered to be binders). Peptides predicted to be MHC-binders are represented by coloured blocks, peptides predicted to be non-MHC binders are represented by grey blocks; the size of the blocks is proportional to the number of peptides in the respective datasets.

**Figure 7 vaccines-10-01907-f007:**
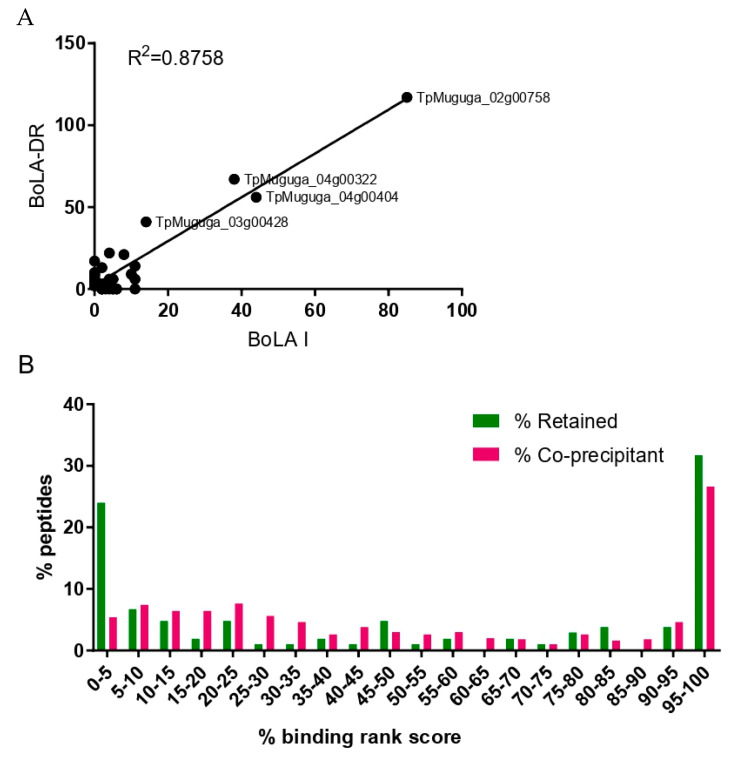
Coprecipitating peptides in the BoLA-MHCI and BoLA-DR datasets. (**A**) A scatterplot showing the level of correlation between the proteins from which coprecipitating peptides were derived in the BoLA-I and BoLA-DR datasets. Each point represents a single protein, and the number of coprecipitating peptides from individual proteins in the BoLA-MHCI and BoLA-DR datasets is shown on the horizontal and vertical axis, respectively. The R^2^ value calculated from the Pearson’s coefficient of correlation is shown. (**B**) The distribution of percent rank prediction scores for peptides considered to be coprecipitants and peptides retained after removal of the coprecipitant peptides in the combined BoLA-DR dataset. Peptides with a percent rank prediction score of <5% are considered to be binders. A small number of short peptides (<9 amino acids long, *n* = 26) that did not receive a percent rank prediction score were ascribed a default value of 100%.

**Table 1 vaccines-10-01907-t001:** *T. parva* peptides derived from 1011TP, 2229TP, and 641TP cell lines. For each peptide, the amino acid sequence (column A), −10 lgP score (B), peptide length (C), sample (D), number of spectra (E), BoLA-I allele to which the peptide has the highest percent rank binding score (F), core peptide sequence (G), percent rank binding prediction score (H), accession number of the source protein (I), description of the source protein (J), the peptide location within the source protein (K and L), and the results of spectral matching of synthetic peptides (M—identified as either positive (matching), negative (not-matching), or ND (not tested)) are shown.

Peptide	−10lgP	Length	Sample	#Spec	Allele	Icore	%Rank	Accession	Protein Description	Peptide Location Start	Peptide Location Stop	Spectral Match
VANTKIEFPEI	31.7	11	641TP	4	6:01301	VANTKIEFPEI	7.15	TpMuguga_01g00263	hypothetical protein	2371	2381	ND
FSVPNQVKAAKVEATIPSHLEKKVITNKKN	73.1	30	1011TP	1	3:00201	FSVPNQVKAAK	92.00	TpMuguga_01g00293	60S ribosomal protein L38, putative	50	79	ND
QAYQQKVDL	21.3	9	2229TP	1	2:02501	QAYQQKVDL	0.45	TpMuguga_01g00386	hypothetical protein	1588	1596	ND
SSISSSLLSVK	23.7	11	1011TP	2	2:01201	SSISSSLLSVK	0.05	TpMuguga_01g00421	hypothetical protein, conserved	365	375	ND
RMYGKGKGISSSSIP	46.8	15	641TP	3	6:01301	RMYGKGKGISSSSI	19.53	TpMuguga_01g00502	40S ribosomal protein S13, putative	3	17	ND
SEKDSYLSSIKKLNL	59.1	15	641TP	1	6:01301	SEKDSYLSSIKKLNL	3.93	TpMuguga_01g00946	ribosome biogenesis regulatory protein, putative	221	235	ND
GKFKNPTKTHKMDEVSESSLQ	28.6	21	2229TP	1	1:02301	GKFKNPTKTH	90.00	TpMuguga_01g01108	hypothetical protein	2	22	ND
GKFKNPTKTHKMDEVSESSLQQK	54.9	23	2229TP	1	1:02301	GKFKNPTKTH	91.67	TpMuguga_01g01108	hypothetical protein	2	24	ND
LNIGRIELIDYI	27.8	12	641TP	6	6:01301	LNIGRIELIDYI	26.37	TpMuguga_01g01149	hypothetical protein	914	925	ND
RALFDDLHRY	22.9	10	2229TP	1	4:02401	RALFDDLHRY	0.28	TpMuguga_01g01198	hypothetical protein	46	55	ND
ILGFDFNKLGKI	28	12	641TP	2	6:01301	ILGFDFNKLGKI	7.89	TpMuguga_02g00219	hypothetical protein	495	506	ND
SLQLKFAQGSDLPNL	51	15	641TP	1	6:01301	SLQLKFAQGSDLPNL	5.77	TpMuguga_02g00321	hypothetical protein	188	202	ND
RVRKLCEYAI	22.2	10	641TP	3	6:01301	RVRKLCEYAI	2.40	TpMuguga_02g00476	crooked neck protein, putative	499	508	ND
ERLAALALYGDYGEFDRKTKEDSK	64.6	24	2229TP	1	6:04001	ERLAALALY	95.00	TpMuguga_02g00758	hypothetical protein	552	575	ND
ERLAALALYGDYGEFD	61	16	641TP	1	6:01301	ERLAALALYGDYGEF	100.0	TpMuguga_02g00758	hypothetical protein	552	567	ND
DRKLFSTKRPSLSL	20.4	14	2229TP	1	6:04001	RKLFSTKRPSLSL	24.46	TpMuguga_02g00803	60S ribosomal protein L18, putative	178	191	ND
QPSYLSQAL	26.7	9	1011TP	1	3:00201	QPSYLSQAL	7.04	TpMuguga_03g00202	DNA polymerase alpha, putative	159	167	ND
SKVDRVSL	23.6	8	2229TP	6	1:02301	SKVDRVSL	0.09	TpMuguga_03g00507	tRNA nucleotidyltransferase (putative)	495	502	ND
SMKGKHELTL	22.1	10	641TP	3	6:01301	SMKGKHELTL	0.01	TpMuguga_03g00747	ATP-dependent RNA helicase, putative	2223	2232	ND
SSIDVNVKL	21.1	9	1011TP	1	3:00201	SSIDVNVKL	0.09	TpMuguga_03g02030	N-terminal region of Chorein, a TM vesicle-mediated sorter family protein	3510	3518	ND
KQVVRDAMVEQDML	32	14	641TP	1	6:01301	KQVVRDAMVEQDML	1.58	TpMuguga_03g02350	RF-1 domain protein	534	547	ND
LELIRARNEI	23.6	10	641TP	1	6:01301	LELIRARNEI	6.82	TpMuguga_04g00031	26S proteasome aaa-ATPase subunit Rpt3, putative	47	56	ND
SPDQPDQHHQPTPAAQP	43.3	17	1011TP	1	3:00201	SPDQPDQHHQPTPAAQP	61.79	TpMuguga_04g00051	polymorphic immunodominant molecule	228	244	ND
FRNEKDLGF	29.1	9	641TP	7	6:01301	FRNEKDLGF	7.44	TpMuguga_04g00227	hypothetical protein	1403	1411	ND
VKKRVHKGKKKARSETYSTYIF	28.5	22	1011TP	1	3:00201	KARSETYSTYIF	96.88	TpMuguga_04g00404	histone H2B-III, putative	2	23	ND
ENKLVEEALK	34.4	10	641TP	2	6:01301	ENKLVEEAL	50.94	TpMuguga_04g00505	hypothetical protein	99	108	ND
KLLYVLKPFI	22.1	10	641TP	2	6:01301	KLLYVLKPFI	4.10	TpMuguga_04g00611	hypothetical protein	234	243	ND
LSPIDILDVAGLVT	21.1	14	1011TP	2	3:00201	LSPIDILDVAGLVT	61.30	TpMuguga_04g00621	DNA repair exonuclease, putative	401	414	ND
IGSAIKDNPAFITL	46.3	14	2229TP	2	6:04001	IGSAIKDNPAFITL	1.66	TpMuguga_01g00188	prohibitin, putative	231	244	Positive
RMDDKSGGLL	42.7	10	641TP	1	6:01301	RMDDKSGGLL	1.20	TpMuguga_01g00736	hypothetical protein	20	29	Positive
GEFEKKYIPTL	50.7	11	641TP	1	6:01301	GEFEKKYIPTL	0.15	TpMuguga_01g00757	GTP-binding nuclear protein ran, putative	30	40	Positive
VQHIPVDDFSGLQTEVVANE	63.5	20	1011TP	6	3:00201	VQHIPVDDFSGLQTEV	97.35	TpMuguga_01g00924	60S ribosomal protein L31, putative	99	118	Positive
VQHIPVDDFSGLQTEVVANE	57.3	20	2229TP	3	6:04001	VQHIPVDDFSGL	95.77	TpMuguga_01g00924	60S ribosomal protein L31, putative	99	118	Positive
RQMQVKLNLP	23.8	10	641TP	2	6:01301	RQMQVKLNL	0.79	TpMuguga_01g00980	40S ribosomal protein S26e, putative	101	110	Positive
TQYERIKERL	28.8	10	641TP	1	6:01301	TQYERIKERL	0.01	TpMuguga_01g01210	hypothetical protein	2	11	Positive
GQIQDVFKRL	33.1	10	641TP	1	6:01301	GQIQDVFKRL	0.25	TpMuguga_02g00123	RNA helicase-1, putative	189	198	Positive
SKDEHKKLY	39.5	9	2229TP	3	1:02301	SKDEHKKLY	0.07	TpMuguga_02g00142	hypothetical protein	91	99	Positive
ATIIGFHK	28	8	1011TP	1	2:01201	ATIIGFHK	0.11	TpMuguga_02g00222	40S ribosomal protein S29, putative	45	52	Positive
SLKSALIDT	21.5	9	641TP	1	6:01301	SLKSALIDT	22.80	TpMuguga_02g00333	translation initiation factor 6, putative	239	247	Positive
SLKSALIDTLI	25.5	11	641TP	4	6:01301	SLKSALIDTLI	2.47	TpMuguga_02g00333	translation initiation factor 6, putative	239	249	Positive
EIKERLAALAL	35.5	11	1011TP	5	3:00201	EIKERLAALAL	14.51	TpMuguga_02g00758	hypothetical protein	549	559	Positive
EIKERLAALAL	27	11	2229TP	2	6:04001	EIKERLAALAL	14.92	TpMuguga_02g00758	hypothetical protein	549	559	Positive
EIKERLAALALYGDYGEFDRKT	66.9	22	1011TP	3	3:00201	EIKERLAALAL	98.82	TpMuguga_02g00758	hypothetical protein	549	570	Positive
EIKERLAALALYGDYGEFDRKT	73.5	22	2229TP	3	6:04001	EIKERLAALALALY	95.00	TpMuguga_02g00758	hypothetical protein	549	570	Positive
EIKERLAALALYGDYGEFDRKT	72.2	22	641TP	4	6:01301	EIKERLAALAL	95.00	TpMuguga_02g00758	hypothetical protein	549	570	Positive
ERLAALALYGDYGEFDRKT	71.2	19	1011TP	3	3:00201	RLAALALYGDYGEFDRK	93.50	TpMuguga_02g00758	hypothetical protein	552	570	Positive
ERLAALALYGDYGEFDRKT	67.9	19	2229TP	5	6:04001	ERLAALALYGDYGEFDRKT	95.00	TpMuguga_02g00758	hypothetical protein	552	570	Positive
ERLAALALYGDYGEFDRKT	72.2	19	641TP	2	6:01301	ERLAALALYGDYGEFDRKT	100.00	TpMuguga_02g00758	hypothetical protein	552	570	Positive
ERLAALALYGDYGEFDRKTK	61.9	20	2229TP	1	6:04001	RLAALALYGDYGEFDRKTK	92.50	TpMuguga_02g00758	hypothetical protein	552	571	Positive
ERLAALALYGDYGEFDRKTKE	58.1	21	2229TP	1	6:04001	ERLAALALY	95.00	TpMuguga_02g00758	hypothetical protein	552	572	Positive
YGDYGEFDRKT	52.8	11	1011TP	3	3:00201	YGDYGEFDRK	16.16	TpMuguga_02g00758	hypothetical protein	560	570	Positive
YGDYGEFDRKTK	47.7	12	1011TP	2	2:01201	YGDYGEFDRKTK	13.40	TpMuguga_02g00758	hypothetical protein	560	571	Positive
YGDYGEFDRKTKEDSK	53.7	16	1011TP	3	2:01201	YGDYGEFDRKTKEDSK	36.34	TpMuguga_02g00758	hypothetical protein	560	575	Positive
YGDYGEFDRKTKEDSK	37.9	16	2229TP	1	6:04001	YGDYGEFDRKTKEDSK	81.67	TpMuguga_02g00758	hypothetical protein	560	575	Positive
YGDYGEFDRKTKEDSKN	36.3	17	1011TP	1	3:00201	YGDYGEFDRKTKEDSK	94.00	TpMuguga_02g00758	hypothetical protein	560	576	Positive
AKFPGMKKSKGPKDK	54.3	15	2229TP	3	1:02301	AKFPGMKKSKGPKDK	39.77	TpMuguga_02g00895	hypothetical protein	67	81	Positive
FRDDLGSSFTSGYTK	59.3	15	1011TP	2	2:01201	FRDDLGSSFTSGYTK	2.96	TpMuguga_02g00895	hypothetical protein	48	62	Positive
RDDLGSSFTSGYTK	59.9	14	1011TP	1	2:01201	RDDLGSSFTSGYTK	0.58	TpMuguga_02g00895	hypothetical protein	49	62	Positive
SSFTSGYTK	38.4	9	1011TP	1	2:01201	SSFTSGYTK	0.01	TpMuguga_02g00895	hypothetical protein	54	62	Positive
SSFTSGYTKQDLDAKFPGMK	68.6	20	1011TP	5	2:01201	SSFTSGYTK	25.30	TpMuguga_02g00895	hypothetical protein	54	73	Positive
KTAPVTGGVK	20.4	10	1011TP	1	2:01201	KTAPVTGGVK	0.46	TpMuguga_03g00152: TpMuguga_04g00321	histone H3, putative	28	37	Negative
KTAPVTGGVKK	22.7	11	1011TP	4	2:01201	KTAPVTGGVKK	0.16	TpMuguga_03g00152: TpMuguga_04g00321	histone H3, putative	28	38	Negative
SGWEGRTL	32.1	8	2229TP	1	6:04001	SGWEGRTL	0.21	TpMuguga_03g00428	40S ribosomal protein S28e, putative	33	40	Positive
SGWEGRTLI	30.8	9	1011TP	1	3:00201	SGWEGRTLI	0.67	TpMuguga_03g00428	40S ribosomal protein S28e, putative	33	41	Positive
ILRTIVQQL	32.7	9	641TP	3	6:01301	ILRTIVQQL	0.10	TpMuguga_03g00716	40S ribosomal protein S19, putative	121	129	Positive
NSFVTDTFEKL	43.5	11	1011TP	1	3:00201	NSFVTDTFEKL	1.68	TpMuguga_04g00404	histone H2B-III, putative	44	54	Positive
SETYSTYIFKVLK	47.8	13	1011TP	2	2:01201	SETYSTYIFKVLK	3.65	TpMuguga_04g00404	histone H2B-III, putative	15	27	Positive
RLFNFATKRI	37.6	10	641TP	1	6:01301	RLFNFATKRI	2.32	TpMuguga_04g00503	hypothetical protein	236	245	Positive

**Table 2 vaccines-10-01907-t002:** Summary of in vitro determined peptide-MHCI dissociation rates (expressed as half-life of peptide-BoLA-I complexes at 37 °C in hours) for a subset of peptides identified from 1011TP, 2229TP, and 641TP cell lines. The dissociation rate of each peptide binding to BoLA-I alleles 1*02301, 2*01201, 3*00201, and 6*01301 was assayed by a scintillation proximity assay, essentially as previously described (see Materials and Methods). Peptides that managed to form a complex BoLA-I molecule and for which a half-life could be measured were considered to be BoLA-I binders. Scores from the in vitro assay that corroborated the in silico predicted ability of peptides to bind to BoLA-I molecules expressed in the cell lines from which they were identified are highlighted with a dark grey background. Scores that are discrepant with the in silico predictions are highlighted with a black background in white script. Three peptides that bound to 6*01301 (BoLA-A18) but were identified from 1011Tp (BoLA-A10) are shown in light grey script. The scores of negative controls (no peptide) and positive control peptides are shown (no peptide) and positive control peptides are shown.

Peptide	−10lgP	Peptide Length	Sample	Accession	Allele	Core	%Rank	SpectralMatch	1*02301	2*01201	3*00201	6*01301
TQYERIKERL	28.82	10	641TP	TpMuguga_01g01210	6:01301	TQYERIKERL	0.01	positive	0.00	0.00	0.00	36.54
SSFTSGYTK	38.38	9	1011TP	TpMuguga_02g00895	2:01201	SSFTSGYTK	0.01	positive	0.00	1.43	0.00	0.22
SKDEHKKLY	39.48	9	2229TP	TpMuguga_02g00142	1:02301	SKDEHKKLY	0.07	positive	31.74	0.00	0.00	0.00
ILRTIVQQL	32.71	9	641TP	TpMuguga_03g00716	6:01301	ILRTIVQQL	0.10	positive	0.00	0.00	0.00	24.03
ATIIGFHK	27.98	8	1011TP	TpMuguga_02g00222	2:01201	ATIIGFHK	0.11	positive	0.00	0.76	0.00	0.76
GEFEKKYIPTL	50.67	11	641TP	TpMuguga_01g00757	6:01301	GEFEKKYIPTL	0.15	positive	0.00	0.00	0.00	24.56
KTAPVTGGVKK	22.73	11	1011TP	TpMuguga_03g00152:TpMuguga_04g00321	2:01201	KTAPVTGGVKK	0.16	negative	0.00	0.32	0.00	0.48
GQIQDVFKRL	33.14	10	641TP	TpMuguga_02g00123	6:01301	GQIQDVFKRL	0.25	positive	0.00	0.00	0.00	19.10
KTAPVTGGVK	20.38	10	1011TP	TpMuguga_03g00152:TpMuguga_04g00321	2:01201	KTAPVTGGVK	0.46	negative	0.00	0.32	0.00	0.00
SGWEGRTLI	30.79	9	1011TP	TpMuguga_03g00428	3:00201	SGWEGRTLI	0.67	positive	0.00	0.00	0.30	0.00
RQMQVKLNLP	23.76	10	641TP	TpMuguga_01g00980	6:01301	RQMQVKLNL	0.79	positive	0.00	0.00	0.00	10.25
RMDDKSGGLL	42.72	10	641TP	TpMuguga_01g00736	6:01301	RMDDKSGGLL	1.20	positive	0.00	0.00	0.00	11.75
NSFVTDTFEKL	43.45	11	1011TP	TpMuguga_04g00404	3:00201	NSFVTDTFEKL	1.68	positive	0.00	0.00	0.00	0.00
RLFNFATKRI	37.63	10	641TP	TpMuguga_04g00503	6:01301	RLFNFATKRI	2.32	positive	0.00	0.00	0.00	6.76
SLKSALIDTLI	25.47	11	641TP	TpMuguga_02g00333	6:01301	SLKSALIDTLI	2.47	positive	0.00	0.00	0.00	7.89
YGDYGEFDRKTK	47.74	12	1011TP	TpMuguga_02g00758	2:01201	YGDYGEFDRKTK	13.40	positive	0.00	0.10	0.00	0.00
EIKERLAALAL	35.54	11	1011TP	TpMuguga_02g00758	3:00201	EIKERLAALAL	14.51	positive	0.00	0.00	0.00	0.00
YGDYGEFDRKT	52.75	11	1011TP	TpMuguga_02g00758	3:00201	YGDYGEFDRK	16.16	positive	0.00	0.00	0.00	0.00
SLKSALIDT	21.52	9	641TP	TpMuguga_02g00333	6:01301	SLKSALIDT	22.80	positive	0.00	0.00	0.00	0.00
Negative control (no peptide)									0.00	0.00	0.00	0.00
Positive control									3.3	2.00	4.50	7.30

**Table 3 vaccines-10-01907-t003:** Predicted BoLA-I presented *T. parva* peptides identified in this study. For each peptide, the accession number and description of the protein from which it is derived are shown in columns A and B, and specific comments about particular proteins are given in column C. Columns D–H provide details about the individual peptides, including their sequence, length, the sample(s) they were identified in, the BoLA-MHCI allele predicted to present the peptide, and the predicted percent rank binding score.

Accession	Protein Description	Comments	Peptide	Length	Sample	BoLA_MHCI Allele	%Rank
TpMuguga_01g00075	hypothetical protein		NQPKNVVEF	9	5350TP_rpt	3:02701	0.04
TpMuguga_01g00100	Protein arginine N-methyltransferase 5		SEIDVKDVL	9	2824TP_rpt	6:01401	0.01
TpMuguga_01g00151	Insulinase (Peptidase family M16) family protein		SEVAVSAMGPL	11	2824TP_rpt	6:01401	1.87
TpMuguga_01g00176	putative integral membrane protein		VEDEAAYHVQL	11	2824_TP	2:01601	0.03
TpMuguga_01g00188	Tp6—Prohibitin-2		IGSAIKDNPAFITL	14	2229TP	6:04001	1.66
TpMuguga_01g00235	Eukaryotic translation initiation factor 3 subunit A		AEKEIVELV	9	2123TP_rpt	1:01901	0.01
TpMuguga_01g00386	DNA polymerase family A family protein		QAYQQKVDL	9	2229TP	2:02501	0.45
TpMuguga_01g00421	RWD domain protein	Identifications in A10 and A18 samples	SSISSSLLSVK	11	1011TP	2:01201	0.05
TpMuguga_01g00421	RWD domain protein	Identifications in A10 and A18 samples	KLIWRFIRHL	10	641TP_rpt	6:01301	0.29
TpMuguga_01g00461	Cytokine-induced anti-apoptosis inhibitor 1, Fe-S biogenesis family protein		GQVTKASFFSSL	12	641TP_rpt	6:01301	0.23
TpMuguga_01g00471	Bifunctional thioredoxin reductase/thioredoxin	Identification in three different A15 samples	FEYEFPINH	9	2123TP_rpt/2408TP/2408_TP_rpt	1:00901	0.03
TpMuguga_01g00566	Brix domain protein		NKKRPISIGF	10	5350TP_rpt	2:02601	0.13
TpMuguga_01g00736	hypothetical protein	Identification in duplicate A18 samples	RMDDKSGGLL	10	641TP	6:01301	1.20
TpMuguga_01g00757	Ras family protein	Identification in duplicate A18 samples	GEFEKKYIPTL	11	641TP	6:01301	0.15
TpMuguga_01g00808	Ubiquitin elongating factor core family protein		SKKDLFIQF	9	5350TP_rpt	3:02701	0.01
TpMuguga_01g00926	Cofilin/tropomyosin-type actin-binding family protein		VEDHDEVRGALA	12	2824TP_rpt	2:01601	1.95
TpMuguga_01g00934	Heat shock cognate 90 kDa protein		SQFVKYPIQL	10	641TP_rpt	6:01301	0.00
TpMuguga_01g01188	Translation initiation factor IF-2		NNPIGRVGF	9	5350TP_rpt	3:02701	0.10
TpMuguga_01g01198	hypothetical protein		RALFDDLHRY	10	2229TP	4:02401	0.28
TpMuguga_01g01207	Pre-mRNA-splicing factor srp2		FGPINRIDF	9	5350TP_rpt	3:02701	0.90
TpMuguga_01g01210	Mitochondrial carrier family protein		TQYERIKERL	10	641TP	6:01301	0.01
TpMuguga_01g02005	Rab-GTPase-TBC domain protein		KLNEQKILSL	10	641TP_rpt	6:01301	0.08
TpMuguga_01g02030	hypothetical protein	Identifications in A12 and A15 samples	EEIAHVLHY	9	2123TP_rpt	1:01901/1:00901	0.39/1.00
TpMuguga_01g02175	Sas10 C-terminal domain protein		YLHEFHNFI	9	2123TP_rpt	2:02501	1.69
TpMuguga_02g00113	DNA replication licensing factor MCM6		SSLLKLTNK	9	A10TP_rpt	2:01201	0.01
TpMuguga_02g00123	DEAD/DEAH box helicase		GQIQDVFKRL	10	641TP	6:01301	0.25
TpMuguga_02g00142	Polyubiquitin		SKDEHKKLY	9	2229TP	1:02301	0.07
TpMuguga_02g00248	Nucleolar GTP-binding protein 1		HMFSGKRTL	9	641TP_rpt	6:01301	0.05
TpMuguga_02g00488	ATP synthase subunit D		AEDFKSLVI	9	2824TP_rpt	2:01601	0.01
TpMuguga_02g00543	putative integral membrane protein		KLANSKNVSL	10	641TP_rpt	6:01301	0.11
TpMuguga_02g00551	23 kDa piroplasm membrane protein		SKATDRLVV	9	5350TP_rpt	3:02701	0.22
TpMuguga_02g00613	DEAD/DEAH box helicase		AKKITELGF	9	5350TP_rpt	2:02601	0.06
TpMuguga_02g00703	putative integral membrane protein	three different identifications in one A20 sample	VKKLKESLL	9	5350TP_rpt	2:02601	0.02
TpMuguga_02g00703	putative integral membrane protein	three different identifications in one A20 sample	NKLGDPLTL	9	5350TP_rpt	3:02701	0.12
TpMuguga_02g00703	putative integral membrane protein	three different identifications in one A20 sample	YKPEGMEYPF	10	5350TP_rpt	3:02701	1.59
TpMuguga_02g00706	hypothetical protein		YQKNSNNPFM	10	5350TP_rpt	2:02601	1.11
TpMuguga_02g00718	hypothetical protein		GKNSVLLQV	9	5350TP_rpt	2:02601	0.15
TpMuguga_02g00723	hypothetical protein		RTFNDVSKRKH	11	2408_TP	1:00901	0.71
TpMuguga_02g00753	chaperone protein DnaK		TQVGIKVY	8	2408_TP	1:00901	0.15
TpMuguga_02g00895	Tp9—Hypothetical protein	two overlapping identifications in one A10 sample	(RDDLG)SSFTSGYTK	9/14	1011TP	2:01201	0.01/0.58
TpMuguga_02g00896	hypothetical protein		AQGDPVFL	8	2408_TP	2:02501	0.16
TpMuguga_03g00253	hypothetical protein	Identifications in A10, A15 and A20 samples	LQSEVFPNY	9	2408_TP	1:00901	0.11
TpMuguga_03g00253	hypothetical protein	Identifications in A10, A15, and A20 samples	FNFSESKLTF	10	5350_TP	3:02701	1.38
TpMuguga_03g00253	hypothetical protein	Identifications in A10, A15, and A20 samples	LNTSIGGSL	9	A10_TP	3:00201	1.00
TpMuguga_03g00257	hypothetical protein		SQNNRSEMSNL	11	641TP_rpt	6:01301	0.07
TpMuguga_03g00330	hypothetical protein		SSMRDALNPPPTH	13	2408_TP	1:00901	1.16
TpMuguga_03g00388	hypothetical protein		SQKRKNKPL	9	641TP_rpt	6:01301	0.01
TpMuguga_03g00469	High mobility group protein homolog NHP1		AKKDPNAPKRAL	12	5350TP_rpt	2:02601	0.13
TpMuguga_03g00478	DEAD/DEAH box helicase family protein		YVGKAPTLW	9	5350TP_rpt	3:02701	1.32
TpMuguga_03g00507	CCA tRNA nucleotidyltransferase mitochondrial		SKVDRVSL	8	2229TP	1:02301	0.09
TpMuguga_03g00544	putative integral membrane protein		NVFPLILGK	9	A10_TP	2:01201	0.08
TpMuguga_03g00577	Cwf15/Cwc15 cell cycle control family protein		SQQPPSFLNDAVRTDFH	17	2408_TP	1:00901	0.52
TpMuguga_03g00655	N4—Hypothetical protein		GVDVDQLLH	9	2123TP_rpt	1:00901	0.36
TpMuguga_03g00747	DEAD/DEAH box helicase		SMKGKHELTL	10	641TP	6:01301	0.01
TpMuguga_03g00852	Protein transport protein Sec61 subunit alpha	two different identifications in one A10 sample	SSMVMQLLAGSK	12	A10TP_rpt	2:01201	1.27
TpMuguga_03g00852	Protein transport protein Sec61 subunit alpha	two different identifications in one A10 sample	KGTEFEGALISL	12	A10TP_rpt	3:00201	1.43
TpMuguga_03g00858	Multiprotein-bridging factor 1c	Identifications in duplicate A10 samples	AGVELDTQKKFL	12	A10_TP	3:00201	1.36
TpMuguga_03g00861	p150—Hypothetical protein		LGPILIYEDL	10	A10TP_rpt	3:00201	0.10
TpMuguga_03g02030	N-terminal region of Chorein, a TM vesicle-mediated sorter family protein		SSIDVNVKL	9	1011TP	3:00201	0.09
TpMuguga_03g02350	RF-1 domain protein		KQVVRDAMVEQDML	14	641TP	6:01301	1.58
TpMuguga_03g02680	putative integral membrane protein		FGIPLVTK	8	A10TP_rpt	2:01201	0.38
TpMuguga_04g00144	hypothetical protein		NSNELKDIK	9	A10TP_rpt	2:01201	1.60
TpMuguga_04g00206	hypothetical protein		IEDEVCKVI	9	2824TP_rpt	2:01601	0.09
TpMuguga_04g00229	Pescadillo N-terminus family protein		SLMPKKHKRLL	11	641TP_rpt	6:01301	0.08
TpMuguga_04g00233	Protein IWS1 homolog		SQMSRNIESKH	11	2123TP_rpt	1:00901	0.03
TpMuguga_04g00368	26S proteasome non-ATPase regulatory subunit 4 homolog		IGLIASAKL	9	A10TP_rpt	3:00201	0.27
TpMuguga_04g00439	hypothetical protein		NSQLRQKIRSM	11	641TP_rpt	6:01301	1.09
TpMuguga_04g00484	OST-HTH Associated domain protein		SGINLGNVNSL	11	A10_TP	3:00201	0.09
TpMuguga_04g00662	putative integral membrane protein		DKKALTVAL	9	5350TP_rpt	3:02701	0.01
TpMuguga_04g00754	hypothetical protein		SQFPRNPVDSLL	12	641TP_rpt	6:01301	0.01
TpMuguga_04g00785	Helicase C-terminal domain		NNFNHSLL	8	5350TP_rpt	3:02701	0.28
TpMuguga_04g00790	AP2-coincident C-terminal family protein		SQSEEIEKYLH	11	2408_TP	1:00901	0.07
TpMuguga_04g02060	hypothetical protein		HQQDSQYYLTQH	12	2408_TP	1:00901	0.10
TpMuguga_04g02625	Importin subunit alpha-6		KQLRNENEI	9	641TP_rpt	6:01301	0.41

**Table 4 vaccines-10-01907-t004:** Predicted BoLA-DR presented *T. parva* peptides identified in this study. For each peptide, the accession number and description of the protein from which it is derived are shown in columns A and B, and specific comments about particular proteins are given in column C. Column D–H provide details about the individual peptides, including its sequence, length, the sample it was identified in, the BoLA-DR allele predicted to present the peptide and the predicted percent rank binding score (%RPS).

Accession	Protein Description	Comments	Peptide	Length	Sample	BoLA_DR Allele	%RPS
TpMuguga_01g00016	Vacuolar protein sorting/targeting protein 10		LDEKSYTILDTSEGAVI	17	641TP_DR	DRB3_2002	0.23
TpMuguga_01g00324	Tubulin/FtsZ family, GTPase domain protein	four nested peptides in one sample	(EFQ)TNLVPYPRIHFML	13–16	5072TP_DR	DRB3_1001	0.1/0.10/0.15/0.34
TpMuguga_01g00552	hypothetical protein		RAKEYNFISKIVYRS	15	495TP_DR	DRB3_1001	0.06
TpMuguga_01g00701	Rhoptry-associated protein 1 (RAP-1) family protein	two nested peptides in one sample	(SE)VQHVVFSFLNDPYK	14, 16	5350TP_DR	DRB3_1201	0.01/0.03
TpMuguga_01g00937	Cell division control protein 48 homolog E	two nested peptides in one sample	(RPG)RLDQLIYIPLPDLPAR	16, 19	641TP_DR	DRB3_2002	0.00/0.01
TpMuguga_01g00972	hypothetical protein		LPVWEAVNDERVDEA	15	2824TP_DR	DRB3_1601	0.04
TpMuguga_01g00987	P104		VAPKDTTLEYLKVFLNK	17	641TP_DR	DRB3_2002	0.46
TpMuguga_01g01056	p32—Merozoite Antigen	two nested peptides in one sample	SEVKFETYYDDVLFKGK(S)	17, 18	5350TP_DR	DRB3_1201	0.00/0.00
TpMuguga_01g01129	hypothetical protein		TKTVSFSNKISFHYF	15	495TP_DR	DRB3_1001	4.02
TpMuguga_02g00253	ATP synthase, Delta/Epsilon chain, beta-sandwich domain protein		NKDLVFSLLSSHEALY	16	641TP_DR	DRB3_2002	0.06
TpMuguga_02g00551	23 kDa piroplasm membrane protein		EDRLATYKPFTEDPSKKR	18	5350TP_DR	DRB3_1201	0.01
TpMuguga_02g00753	chaperone protein DnaK		DFDQRILNFLVDEFKK	16	5350TP_DR	DRB3_1201	0.12
TpMuguga_04g00395	26S proteasome non-ATPase regulatory subunit 6		KQGDLLINRIQKLSRIIDM	19	2123TP_DR	DRB3_1501	4.22
TpMuguga_04g00719	2-oxoisovalerate dehydrogenase subunit alpha mitochondrial	two nested peptides in one sample	(E)NTKAYELLPGLFDDV	15, 16	641TP_DR	DRB3_2002	0.01/0.06
TpMuguga_04g02435	ARF guanine-nucleotide exchange factor GNL1	two nested peptides in one sample	KEKDFLADITKELDESQ(S)	17, 18	2824TP_DR	DRB3_1601	0.11/0.84

**Table 5 vaccines-10-01907-t005:** *T. parva* proteins that have been shown to contain epitopes recognised by either CD4^+^ and/or CD8^+^ T-cells and as sources of peptides identified in *T. parva* samples during pMHC elution of either BoLA-I or BoLA-DR. Identity of confirmed CD4^+^ and CD8^+^ T-cell epitopes come from Graham et al. (2006) and Morrison et al. (2021).

Accession	Protein Name	Antigens Recognised by *T. parva*-Specific		Identified in Peptide Elution From	
		CD8	CD4	DR	MHCI
TpMuguga_01g00188	Tp6—Prohibitin-2	CD8	-	-	MHCI
TpMuguga_02g00123	Tp32—DEAD/DEAH box helicase	CD8	CD4	-	MHCI
TpMuguga_02g00895	Tp9—Hypothetical protein	CD8	CD4	-	MHCI
TpMuguga_03g00655	Tp13—Hypothetical protein	-	CD4	-	MHCI
TpMuguga_03g00861	Tp20—Hypothetical protein	-	CD4	-	MHCI

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
