# Peer review of "Immunopeptidomic Analysis of BoLA-I and BoLA-DR Presented Peptides from Theileria parva Infected Cells"

_vaccines, 2022, doi:10.3390/vaccines10111907_

Round 1

Reviewer 1 Report

Brief Summary

The manuscript of Connelley et all describes a very comprehensive study on the immunopeptidome of bovine cells infected with protozoan Theileria parva, the causative agent of east coast fever in cattle. The authors evaluated peptides presented on MHC-I as well as MHC-II on BoLa cells, and looked at three different MHC-I haplotype combinations for MHC-I. The T. parva peptide hits that were retrieved for T. parva were thoroughly analyzed and discussed. Notably, a refining process was introduced where proteins from co-precipitated T. parva peptide hits were excluded from the final dataset. This resulted in a dataset of 74 and 15 T. parva MHC-I and MHC-II derived peptides, respectively.

Broad comments

This manuscript is very well drafted, the story the authors are trying to tell is clear.  I enjoyed readying the manuscript. My only broad question is about the exclusion of co-precipitated parasite proteins. Since this exclusion is based on the criteria that the peptides have a poor % rank binding prediction score determined by netMHCpan, what if the prediction of netMHCpan is incorrect? Could it be that you would remove proteins that have peptides that are actually binding to MHC, but are not ranked correctly by netMHCpan?

Specific comments

B. taurus and T.parva should be written in italic throughout the document, even in the figure legends. Please correct.

Table II: peptide length is displayed in two separate columns, while showing the same information. I believe one column can be removed.

Author Response

We thank the reviewer for the positive comments.

With respect to the broad comment made, the reviewer is correct in that we can’t be absolutely confident that all of the peptides removed as co-precipitants are not genuine BoLA-binding peptides. However, the study demonstrates that i) the vast majority of peptide with poor predicted binding stem from a very small subset of Theileria parva proteins and that they are found across multiple samples irrespective of the BoLA background; and ii) a subset of such peptides with poor predicted binding fail to form stable pMHC complexes as assayed using in vitro peptide-MHC binding stability assays (whereas almost all peptides with predicted binding where found to form stable complexes). Together this implies strongly that the peptides are BoLA irrelevant co-precipitates. As the purpose of the study was to identify candidate vaccine antigens, the identification of a large number of predicted BoLA binders enabled us to employ a high threshold for exclusion of peptides that appeared to be co-precipitates, with the acknowledgement that some genuine peptides may have been excluded.

With respect to the specific comments:

Bos taurus and Theileria parva have been italicized throughout the manuscript.

Table II has been corrected - thank you for identifying this duplication

Reviewer 2 Report

Dear authors,

This is excellent scientific explanation on BoLA-I and BoLA-DR presented peptide.  Immunopeptidomic analysis of BoLA-I and BoLA-DR pre-2 sented peptides from Theileria parva infected cells.

Why you choose only Theileria parva infected cells for this peptide?

How you know that this peptide stability?

What is the reason to proof that this peptide would be beneficial?

Author Response

We thank the reviewer for the positive comments.

Why you choose only Theileria parva infected cells for this peptide?

Theileria parva was selected as the focus for this study as it was the pathogen that was the main research interest of the consortium undertaking the work.

Theileria is a particularly relevant model pathogen in which to apply immunopeptidomics as i) it is a pathogen of economic significance in the areas where it is present, ii) as a eukaryote it has a large proteome and in the field it exhibits high strain diversity, and iii) it effects cattle populations that are likely to demonstrate a wide diversity of MHC genotypes; consequently there is a pressing need for new vaccines to be developed but antigen identification is particularly challenging and so novel methods that can address this issue are of real value. In terms of practicality the Theileria model also benefits from Theileria-infected cells exhibiting a transformed phenotype in vitro, enabling the large number of infected cells required for immunopeptidomics to be easily and rapidly generated. However, we agree with the reviewer that application of immunopeptidomics to other veterinary pathogens would be of value and we hope that future work will be expanded to accommodate these.

How you know that this peptide stability?

We have directly assessed the stability of a small subset of the BoLA-I eluted peptides in in vitro assays (as reported in Table II). This analysis demonstrated a high level of correlation between the predicted MHC binding ability and in vitro data - suggesting that the algorithm generating the predictions was robust. Supporting this conclusion, the known BoLA-I epitopes from Theileria parva have all been predicted to be strong binders by the algorithm generating these predictions. 

What is the reason to proof that this peptide would be beneficial?

To prove that these peptides are beneficial would require experiments to show that when delivered by a relevant vaccine vehicle they are able to induce T cell responses. These experiments were outwith the current experimental plan but form a core component of future studies that are being planned as follow-on studies.